# POLICY GRADIENT OPTIMIZATION FOR MARKOV DECISION PROCESSES WITH EPISTEMIC UNCERTAINTY AND GENERAL LOSS FUNCTIONS

## ABSTRACT

Motivated by many application problems, we consider Markov decision processes (MDPs) with a general loss function and unknown parameters. To mitigate the epistemic uncertainty associated with unknown parameters, we take a Bayesian approach to estimate the parameters from data and impose a coherent risk functional (with respect to the Bayesian posterior distribution) on the general loss function. Since this formulation usually does not satisfy the interchangeability principle, it does not admit Bellman equations and cannot be solved by approaches based on dynamic programming. Therefore, we develop a policy gradient optimization approach to address this problem. We utilize the dual representation of the coherent risk measure and extend the envelope theorem to derive the policy gradient. Our extension of the envelope theorem from the discrete case to the continuous case may be of independent interest. We then show the convergence of the proposed algorithm with a convergence rate of $\mathcal{O}((1-\epsilon)^t)$, where $t$ is the number of policy gradient iterations and $\epsilon$ is the accuracy. We further extend our algorithm to an episodic setting, and establish the consistency of the extended algorithm and provide bounds on the number of iterations needed to achieve an error bound $\mathcal{O}(\epsilon)$ in each episode.

## 1 INTRODUCTION

Markov decision process (MDP) is a paradigm for modeling sequential decision making under uncertainty, with a primary focus on identifying an optimal policy that minimizes the (discounted) expected total cost. However, the standard form of MDP is not sufficient for modeling some practical problems. For example, consider a self-driving car operating in a dynamic urban environment. On one hand, the self-driving car must not only reach its destination efficiently but also safely against unpredictable events, requiring a general optimization objective within the MDP framework. On the other hand, the car has incomplete knowledge about its environment, such as road conditions. In such a case, the decision maker encounters two key challenges: the need for a general performance measure to address intrinsic uncertainty, and epistemic uncertainty about the environment. This paper is motivated by these challenges and aims to address both the general loss function and epistemic uncertainty simultaneously in the MDP framework.

There is extensive literature addressing general loss functions and epistemic uncertainty separately. For instance, risk-sensitive objectives have been explored in the contexts of MDPs (Howard & Matheson, 1972; Ruszczyński, 2010; Mannor & Tsitsiklis, 2011; Petrik & Subramanian, 2012), stochastic optimal control (Borkar & Meyn, 2002; Moon, 2020), and stochastic programming (Shapiro, 2012; Pichler et al., 2022) literature. These objectives cannot be simply represented as the total expected cost. Epistemic uncertainty arises when some MDP parameters, such as transition probabilities, are unknown and must be estimated from available data. This discrepancy between the estimated and true MDP is referred to as epistemic uncertainty. Numerous approaches have been proposed to address epistemic uncertainty in MDPs, with robust MDP (Nilim & Ghaoui, 2004; Iyengar, 2005; Delage & Mannor, 2010; Wiesemann et al., 2013; Petrik & Russel, 2019) being one of the most widely adopted formulations. A more flexible and less conservative formulation, coined as Bayesian Risk MDP, was recently proposed by Lin et al. (2022).

However, there is no existing literature that addresses both a general loss function and epistemic uncertainty simultaneously. To the best of our knowledge, this paper is the first to consider this problem. In this work, we study MDPs with a general loss function, particularly focusing on loss functions that are convex in terms of the occupancy measure. Additionally, to handle both epistemic uncertainty and intrinsic uncertainty, we take a Bayesian approach to estimate the unknown parameters (such as transition probabilities) with data, and impose a coherent risk functional (with respect to the Bayesian posterior distribution) on the general convex loss function, using a fixed batch of data. Therefore, the problem is framed as an offline optimization task. Our composite objective consists of two components: the outer general coherent risk measure and the inner general convex loss function. To determine the optimal policy for this composite problem, we use a policy optimization approach, which directly optimizes policies and accommodates complex, high-dimensional representations such as neural networks. This method typically employs parameterized policies and utilizes a policy gradient approach, introduced by Sutton et al. (1999), to search for the optimal solution. For the outer layer, the coherent risk measure admits a dual representation as demonstrated by Shapiro et al. (2021), which can be expressed as the supremum of the expectation over a risk envelope set. We extended the envelope theorem in Milgrom & Segal (2002) to obtain the policy gradient. A similar approach was taken by Tamar et al. (2015), but their consideration of a discrete parameter space limits the applicability of their method to our problem. Our extension from the discrete case to the continuous case for the envelope theorem may be of independent interest. The derived policy gradient involves the gradient of the loss function with respect the policy parameter, which can then be estimated by different methods. In particular, we adapt the recent variational approach proposed by Zhang et al. (2020) to construct the gradient estimator. Other methods, such as zeroth-order estimation method proposed by Balasubramanian & Ghadimi (2022), could also be used to estimate the inner gradient. By incorporating the inner gradient estimator into the policy gradient, we derive the gradient estimator for the composed objective and use policy gradient descent to optimize the problem. To make our approach more applicable with new observed data, we further extend our approach to the episodic setting, where the agent iteratively applies the current policy to gather more data and updates the policy based on new environment estimates informed by the additional data.

Our choice of policy gradient method for this problem is not only due to its popularity but also because algorithms based on dynamic programming are not applicable to general loss function that is not in the standard form of cumulative sum. Therefore, our approach is completely different from most robust MDPs or Bayesian risk MDPs which relies on dynamic programming. However, for feasibility of the policy gradient method, we assume the general loss function is convex. The convex loss functions are widely used, as discussed by Pennings & Smidts (2003), and are sufficiently general to encompass many of the previously mentioned examples (e.g., risk-sensitive MDPs and constrained MDPs) as special cases. More discussions about convex RL is offered in Appendix A.1 The standard expected total cost can be viewed as such a special case, where the loss function is a linear function of the occupancy measure. The dynamic programming approach to solving MDPs involves the use of Bellman equations. However, the derivation of Bellman equations relies on the interchangeability principle, which may not hold for general convex loss functions. For a more detailed discussion on why the interchangeability principle fails for general convex loss functions, we refer readers to Rockafellar & Wets (2009) for the expectation operator and Shapiro (2017) for general risk functionals. It is also worth noting that the Bayesian approach has been considered by Duff (2002); Poupart et al. (2006); Abbasi-Yadkori & Szepesvári (2015); Imani et al. (2018); Derman et al. (2020); Lin et al. (2022); Wang & Zhou (2023), where the Bayesian update accounts for future data realization and enables the use of dynamic programming algorithms.

For the composite problem in our proposed formulation, there have been dedicated efforts to solve MDPs with some special objectives using the policy gradient algorithm. For example, Chow & Ghavamzadeh (2014) applied Conditional Value-at-Risk(CVaR) to the total cost and developed policy gradient and actor-critic algorithms, each utilizing a distinct method to estimate the gradient and update policy parameters in the descent direction. In contrast, we consider a broader composition of a general coherent risk measure and a general loss function, allowing more flexible objectives. Note that although the composition of a coherent risk measure and a convex loss function is convex in the occupancy measure, it is generally non-convex in the policy parameters, which introduces additional challenges for our convergence analysis. Finally, the work most relevant to ours is perhaps Zhang et al. (2020), which addresses a reinforcement learning problem with a general convex loss function and derives the variational policy gradient theorem with a global convergence guarantee. However,

our work differs in that we consider an offline planning problem in an MDP with unknown transition probabilities, which are estimated from data. Therefore, we address not only a general convex loss function but also epistemic uncertainty. This introduces additional challenges related to risk measures, and the robustness of the proposed formulation is a key consideration.

Our contributions are summarized as follows: (1) We propose a Bayesian risk formulation for MDPs with a general convex loss function and develop a policy gradient algorithm to solve for the optimal policy. The proposed formulation jointly mitigates both epistemic and intrinsic uncertainty; (2) We extend the envelope theorem to the dual representation of the coherent risk measure, and then apply the envelope theorem to derive the policy gradient. Our extension from the discrete case to the continuous case for the envelope theorem may be of independent interest; (3) We prove the convergence of the proposed algorithm and establish its convergence rate as $\mathcal{O}((1-\epsilon)^t)$, where $t$ is the number of policy gradient iterations and $\epsilon$ is the accuracy; (4) We extend our policy gradient algorithm to the episodic setting, and prove the asymptotic convergence of the episodic minimizer of our Bayesian formulation to a global minimizer of the MDP problem under the true environment. Moreover, we show the number of iterations required in any episode to maintain an optimality gap $\mathcal{O}(\epsilon)$ under our Bayesian formulation.

## 2 PROBLEM FORMULATION

Consider an infinite-horizon Markov Decision Process (MDP) over a finite state space $\mathcal{S}$ and a finite action space $\mathcal{A}$. For each state $s \in \mathcal{S}$ and action $a \in \mathcal{A}$, a transition to the next state $s'$ follows the transition kernel $P^*$, i.e. $s' \sim P^*(\cdot|s,a)$. A stationary policy $\pi$ is defined as a function mapping from the state space to a probability simplex $\Delta(\cdot)$ over the action space. Given any transition probability $P$, define $\lambda^{\pi,P}$ to be the discounted state-action occupancy measure under policy $\pi$:

$$\lambda_{sa}^{\pi,P} = \sum_{t=0}^{\infty} \gamma^t \cdot \mathbb{P}\left(s_t = s, a_t = a \mid \pi, s_0 \sim \tau, P\right), \ \forall (s,a) \in \mathcal{S} \times \mathcal{A}, \tag{1}$$

where $\tau$ is the initial distribution, $\gamma \in (0,1)$ is the discount factor.

As mentioned in introduction, in many application problems such as a self-driving car in a dynamic urban environment, the decision maker faces two kinds of challenges: the epistemic uncertainty about the environment and a general performance measure for the intrinsic uncertainty. In this paper, we aim to address both challenges together. We consider a general loss function $F(\lambda, P)$ defined over the occupancy measure $\lambda$ and transition kernel $P$, which is assumed to be convex in $\lambda$. In practice, the true distribution $P^*$ is usually unknown and needs to be estimated. In this work, we take a Bayesian approach to estimate the environment. We assume that the transition kernel $P^* \equiv P_{\theta^c}$ is parameterized by $\theta^c$, where $\theta^c \in \Theta$ is the true but unknown parameter value, $\Theta \subseteq \mathbb{R}^p$ is the parameter space, and $p$ is the dimension of $\Theta$. Many real-world problems exhibit the characteristic of relying on a parametric assumption. In the example of a self-driving car, the noise in sensor measurements may be assumed to follow an unknown Gaussian distribution.

Under the parametric assumption, we assume we have access to some data which are state transitions $\zeta = (s,a,s')$, where $s'$ follows the distribution $P_{\theta^c}(\cdot|s,a)$ and define $P_{\theta^c}(\zeta) := P_{\theta^c}(s'|s,a)$. Now given a fixed batch of data $\zeta^{(N)}$ of $N$ samples, we can update the posterior distribution (denoted by $\mu_N$) on the parameter $\theta$ using the Bayes rule: $\mu_N(\theta) = \frac{P_\theta(\zeta^{(N)})\mu_0(\theta)}{\int_{\theta'} P_{\theta'}(\zeta^{(N)})\mu_0(\theta')d\theta'}$, where $\mu_0$ is a prior distribution of $\theta$ we assume. Furthermore, as discussed before, model mis-specification caused by the lack of data could lead to sub-optimality of the learned policy when it is implemented in a real-world setting. Hence, we further impose a risk functional on the objective with respect to (w.r.t.) the Bayesian posterior to account for the epistemic uncertainty, which results in the following composed formulation:

$$\min_\pi \rho_{\theta \sim \mu_N}(F(\lambda^{\pi,P_\theta}, P_\theta)) \tag{2}$$

where $\rho$ is a general coherent risk measure [1] w.r.t. the posterior $\mu_N$. We aim to solve problem equation 2 in this paper. Detailed introduction about coherent risk measures can be found in (Artzner

---

[1] Let $(\Omega, \mathcal{F}, \mathbb{P})$ w.r.t. the posterior $\mu_N$ be a probability space and $\mathcal{X}$ be a linear space of $\mathcal{F}$-measurable functions $X : \Omega \to \mathbb{R}$. A risk measure is a function $\rho : \mathcal{X} \to \mathbb{R}$ which assigns to a random variable $X$ a real number representing its risk. A coherent risk measure satisfies properties of monotonicity, sub-additivity, homogeneity, and translational invariance.

et al., 1999). By this formulation, we look for a policy that minimizes a performance measure taking into account to the epistemic uncertainty caused by lack of data for a general convex loss function.

If $F$ is a linear function of $\lambda$, i.e. $F(\lambda, P) = \langle \lambda, c \rangle$ for a cost vector $c \in \mathbb{R}^{|\mathcal{S}| \times |\mathcal{A}|}$, and the posterior $\mu_N$ is a singleton on the true parameter $\theta^c$, then the risk measure just considers the performance on this singleton and equation 2 reduces to the classical MDP problem. Next, we give some examples that are not in the classical form of MDP but fall into our framework. We list one example below, which is motivated by safe reinforcement learning, and more examples can be found in Appendix C.

**Example 1** (Risk-Averse Constrained MDP). *In safe reinformcent learning problems, one usually considers a constrained MDP (Altman, 2021), where the goal is to minimize the total expected discounted cost under a risk-averse constraint. Given a random vector penalty $d$, the risk-averse constraint is to control a risk measure of the total expected discounted penalty. This leads to the following constrained MDP formulation:*

$$\min_{\pi} \mathbb{E}[\sum_{t=0}^{\infty} \gamma^t c(s_t, a_t) \mid \pi, s_0 \sim \tau] \quad s.t. \ \rho\left(\mathbb{E}[\sum_{t=0}^{\infty} \gamma^t d(s_t, a_t) \mid \pi, s_0 \sim \tau]\right) \leq D,$$

*where $\rho$ is a coherent risk measure, such as Conditional Value-at-Risk (CVaR)[2] Using Lagrangian relaxation, we can choose $F$ to be a convex function of $\lambda$, i.e., $F(\lambda, P) = \langle \lambda, c \rangle + \ell(\rho(\langle \lambda, d \rangle) - D)$, where $\ell$ is the Lagrange multiplier.*

## 3 POLICY GRADIENT ALGORITHM: DERIVATION AND ESTIMATION

As discussed in the introduction, the dynamic programming type of algorithms may not be readily available for a general convex loss function $F(\cdot)$. Therefore, we adopt the policy gradient algorithm, which directly optimizes parameterized policies. Consider a stochastic parameterized policy $\pi_\alpha : \mathcal{S} \to \Delta(\mathcal{A})$, parameterized by $\alpha \in W \subset \mathbb{R}^d$. To directly work on the parameterized policy, we denote $F(\lambda^{\pi_\alpha, P_\theta}, P_\theta)$ by $C(\alpha, \theta)$. The policy optimization problem equation 2 then becomes:

$$\min_{\alpha} G(\alpha) := \rho_{\theta \sim \mu_N}(C(\alpha, \theta)). \tag{3}$$

It is worth mentioning that $G(\alpha)$ is not necessarily a convex function though $F$ is convex w.r.t. $\lambda$. However, we can still reach a global minimum of $G(\alpha)$ by the policy gradient descent method (see more detailed discussion in Section 4.2). In the rest of the section, we derive the policy gradient to the proposed formulation equation 3 using the envelope theorem, and construct the policy gradient estimator. It should be noted that our proposed formulation allows for flexible methods to estimate the policy gradient, including the variational approach such as in Zhang et al. (2020), and the zeroth-order method such as in (Balasubramanian & Ghadimi, 2022).

### 3.1 PRELIMINARIES

Note that $\Theta$ equipped the posterior distribution $\mu_N$ is a probability space. To ensure the objective $G(\alpha)$ is well defined, we first make a basic assumption about $C(\alpha, \theta) \in \mathcal{Z} := L_p(\Theta, \mu_N)$.

**Assumption 3.1.** $C(\alpha, \theta) \in \mathcal{Z} = \{f : \int_{\Theta} |f(\theta)|^p d\mu_N(\theta) < \infty\}, \forall \alpha \in W, \text{ for some } p \geq 1$.

The choice of $p$ depends on the specific coherent risk measure. For example, $p$ can be chosen as 1 for CVaR introduced in Example 1. Let $\mathcal{B} := \{\xi \in \mathcal{Z}^* : \int_{\Theta} \xi(\theta)\mu_N(\theta)d\theta = 1, \xi \succeq 0\}$, where $\mathcal{Z}^* := L_q(\Theta, \mu_N)$ is the dual space of $\mathcal{Z}$ with $1/p + 1/q = 1$. It is well known that a coherent risk measure has a dual representation, which is shown in Shapiro et al. (2021).

**Theorem 1.** *(Theorem 6.6 in (Shapiro et al., 2021).) A risk measure $\rho : \mathcal{Z} \to \mathbb{R}$ is coherent if and only if there exists a convex bounded and closed set (also known as risk envelope) $\mathcal{U} = \mathcal{U}(\mu_N) \subset \mathcal{B}$ such that $\rho(Z) = \max_{\xi : \xi \in \mathcal{U}(\mu_N)} \mathbb{E}_\xi[Z]$, where $\mathbb{E}_\xi[Z] := \int_{\theta \in \Theta} Z(\theta)\xi(\theta)\mu_N(\theta)d\theta$.*

Note $\xi$ could be viewed as perturbation on the posterior $\mu_N$ that satisfies certain conditions, and the risk measure can be understood as the extreme performance for these perturbations. Theorem 1 implies that a functional $\rho$ defined by $\rho(Z) = \max_{\xi : \xi \in \mathcal{U}} \mathbb{E}_\xi[Z]$ is a coherent risk measure if $\mathcal{U} \subset \mathcal{B}$ is convex, bounded and closed. In this paper we only focus on a special class of coherent risk measures whose risk envelope can be written under the form in the following.

---

[2]CVaR$(X) = \mathbb{E}[X | X \geq v_\beta(X)]$, where $v_\beta(X)$ is a $\beta$-quantile of $X$, i.e. $\mathbb{P}(X \geq v_\beta(X)) = 1 - \beta$

**Definition 3.1.** *For each given policy parameter $\theta \in \mathbb{R}^K$, there exists an expression for the risk envelope $\mathcal{U}$ of the coherent risk measure $\rho$ in the following form:*

$$\mathcal{U}(\mu_N) = \{\xi \in \mathcal{Z}^* : g_e(\xi, \mu_N) = 0, \forall e \in \mathcal{E}, f_i(\xi, \mu_N) \leq 0, \forall i \in \mathcal{I},$$

$$\int_{\theta \in \Theta} \xi(\theta)\mu_N(\theta)d\theta = 1, \xi(\theta) \geq 0, \|\xi\|_q \leq B_q\},$$

*where constraint $g_e(\xi, \mu_N)$ is an affine function in $\xi$, each constraint $f_i(\xi, \mu_N)$ is a convex function in $\xi$, $\| \cdot \|_q$ is the $L_q$ norm in $\mathcal{Z}^*$, and there exists a strictly feasible point $\xi$. $\mathcal{E}$ and $\mathcal{I}$ here denote the sets of equality and inequality constraints, respectively. Furthermore, for any given $\xi \in \mathcal{B}$, $f_i(\xi, \mu_N)$ and $g_e(\xi, \mu_N)$ are twice differentiable in $\mu_N$, and there exists a $M > 0$ such that*

$$\max\left\{\max_{i \in \mathcal{I}}\left|\frac{df_i(\xi, \mu_N)}{d\mu_N(\theta)}\right|, \max_{e \in \mathcal{E}}\left|\frac{dg_e(\xi, \mu_N)}{d\mu_N(\theta)}\right|\right\} \leq M, \forall \omega \in \Omega.$$

The conditions on $g_e$ and $f_i$ ensure that risk envelope $\mathcal{U}(\mu_N)$ is a convex closed set, and the condition $\|\xi\|_q \leq B_q$ makes $\mathcal{U}(\mu_N)$ bounded. A similar assumption is considered in Tamar et al. (2015) (see their Assumption 2.2). The assumption about bounded derivatives can be easily satisfied if $\Theta$ is compact. A major difference is that Tamar et al. (2015) only consider the case where $\Theta$ is finite, while we extend it to the continuous case, leading to a functional problem over an infinite dimensional space instead of a finite-dimensional case. Therefore, we extend the result in Tamar et al. (2015) to the infinite dimensional space, which is shown in Theorem 2. It should also be noted that the function forms of $g_e(\cdot)$ and $f_i(\cdot)$ can be exactly specified for a given coherent risk measure. We refer the readers to Appendix D for some examples of the envelope set for coherent risk measures. More examples that satisfy Definition 3.1 can be found in Section 6.3.2 (Shapiro et al., 2021), which covers most common coherent risk measures.

## 3.2 DERIVATION OF POLICY GRADIENT

According to Theorem 1, we can write the coherent risk measure as a maximization problem, where the decision variable is $\xi$ and the objective is a linear functional of $\xi$:

$$\rho_{\theta \sim \mu_N}(C(\alpha, \theta)) = \max_{\xi : \xi \in \mathcal{U}(\mu_N)} \mathbb{E}_\xi[C(\alpha, \theta)] = \max_{\xi : \xi \in \mathcal{U}(\mu_N)} \int_{\theta \in \Theta} \xi(\theta)\mu_N(\theta)C(\alpha, \theta)d\theta. \quad (4)$$

For the maximization problem equation 4, we define the Lagrangian function as:

$$L_\alpha(\xi, \lambda^{\mathcal{P}}, \lambda^{\mathcal{E}}, \lambda^{\mathcal{I}}) = \int_{\theta \in \Theta} \xi(\theta)\mu_N(\theta)C(\alpha, \theta)d\theta - \lambda^{\mathcal{P}}\left(\int_{\theta \in \Theta} \xi(\theta)\mu_N(\theta)d\theta - 1\right)$$

$$- \sum_{e \in \mathcal{E}} \lambda^{\mathcal{E}}(e)g_e(\xi, \mu_N) - \sum_{i \in \mathcal{I}} \lambda^{\mathcal{I}}(i)f_i(\xi, \mu_N). \quad (5)$$

Using the Lagrangian relaxation equation 5, we derive the policy gradient for equation 4 in Theorem 2. For this purpose, We need some mild assumptions about continuity on the objective function.

**Assumption 3.2.** *(1) $\nabla_\lambda F(\lambda, P)$ is uniformly bounded by $L_{F,\infty}$ for any $\lambda$ and $P$ w.r.t. $\| \cdot \|_\infty$; (2) $\nabla C(\alpha, \theta)$ is $L_{C,2}$-Lipschitz continuous w.r.t. $\theta \in \Theta$ and $\| \cdot \|_2$ for any $\alpha \in W$; (3) $\nabla C(\alpha, \theta)$ is uniformly bounded by $B$ for any $\alpha \in W$ and $\theta \in \Theta$ w.r.t. $\| \cdot \|_2$; (4) $\Theta \subseteq \mathbb{R}^p$ is compact and convex; (5) $W$, the domain of $\alpha$, is bounded by $B_W$.*

Assumption 3.2 requires the uniform boundedness and Lipschitz continuity of $\nabla C$ and $\nabla F$, where $C(\alpha, \theta) = F(\lambda^{\pi_\alpha, P_\theta}, P_\theta)$. One sufficient condition easy to verify for Assumption 3.2 to hold is: each component in the composed function $F(\lambda^{\pi_\alpha, P_\theta}, P_\theta)$ is (somewhere) twice continuously differentiable w.r.t parameters $\alpha, \theta$, and the domains of two parameters are compact convex sets.

**Theorem 2.** *Assume Assumption 3.1 3.2 hold, and $\rho$ satisfies Definition 3.1. Assume that $\mu_N$ is a Radon measure [3]. Define $\xi^* \in \arg\max_{0 \leq \xi, \|\xi\|_q \leq B_q} \min_{\lambda^{\mathcal{I}} \geq 0, \lambda^{\mathcal{P}}, \lambda^{\varepsilon}} L_\alpha(\xi, \lambda^{\mathcal{P}}, \lambda^{\mathcal{E}}, \lambda^{\mathcal{I}})$. Then we have the policy gradient*

$$g(\alpha) := \nabla_\alpha \rho_{\theta \sim \mu_N}(C(\alpha, \theta)) = \int_{\theta \in \Theta} \xi_\alpha^*(\theta)\mu_N(\theta)\nabla_\alpha C(\alpha, \theta)d\theta. \quad (6)$$

---

[3] $\mu_N$ is a Radon measure on $\Theta$ if (i) $\mu_N(\Theta)$ is finite, (ii) for all Borel set $E \subseteq \Theta$, we have $\mu_N(E) = \inf\{\mu_N(U) : E \subseteq U, U \text{ is open}\}$ and $\mu_N(E) = \sup\{\mu_N(K) : K \subseteq E, K \text{ is compact}\}$. In the case of continuous parameter space $\Theta$, if the prior is a continuous distribution and the likelihood function is continuous in $\theta$, then the posterior is Radon. For discrete case, we don't need to care about this assumption. Thus it hold in most cases that we may care about, and most common probability distributions are Radon Measures.

Proof details of Theorem 2 can be found in Appendix B.1. Theorem 2 implies that we can plug in a saddle point of Lagrangian equation 5 into equation 6 to get the policy gradient. However, equation 6 still involves $\nabla C$, the gradient of the loss function, and the integration w.r.t. the posterior $\mu_N$. In the next subsection, we show how to estimate the policy gradient in equation 6.

### 3.3 CONSTRUCTION OF THE POLICY GRADIENT ESTIMATOR

In this section, we focus on how to estimate the policy gradient $g(\alpha)$ and denote its estimator by $\widehat{g}(\alpha)$. We first need to find $\xi^*$ in Theorem 2. For some coherent risk measures, the closed-form expression of $\xi^*$ is known. Taking CVaR with risk level $\beta \in (0,1)$ as an example, $\xi^*(\theta) = \frac{1}{1-\beta}$ if $C(\alpha, \theta) \geq v_\beta$ and 0 otherwise, where $v_\beta$ is the $\beta$-quantile of $C(\alpha, \theta)$. For a general coherent risk measure, we can use the approach sample average approximation (SAA). We first sample $\theta_k, k = 1, \ldots, r_N$, from $\mu_N$, and then solve the following SAA problem for the solution $\xi^*(\theta_k)$ for each $k$:

$$\max_{\substack{\xi \geq 0, \\ \frac{1}{r_N}\sum_{k=1}^{r_N}|\xi(\theta_k)|^q \leq B_q}} \min_{\lambda^{\mathcal{I}} \geq 0, \lambda^{\mathcal{E}}} \frac{1}{r_N}\sum_{k=1}^{r_N} \xi(\theta_k)C(\alpha, \theta_k) - \lambda^{\mathcal{P}}\left(\frac{1}{r_N}\sum_{k=1}^{r_N}\xi(\theta_k) - 1\right)$$
$$- \sum_{e \in \mathcal{E}} \lambda^{\mathcal{E}}(e)g_e\left(\xi^{(r_N)}, \mu_N(r_N)\right) - \sum_{id \in \mathcal{I}} \lambda^{\mathcal{I}}(k)f_i\left(\xi^{(r_N)}, \mu_N(r_N)\right) \quad (7)$$

Notice the objective in equation 7 is linear w.r.t. $\lambda^{\mathcal{I}}, \lambda^{\mathcal{E}}$ and concave w.r.t $\xi$, and the domain of $\xi$ is a convex bounded set in $\mathbb{R}^{r_N}$. Thus, equation 7 can be solved by any max-min optimization algorithm for a concave-convex function, such as alternating gradient descent ascent. Here we assume that we can solve equation 7 to derive $\xi^*(\theta_k)$ accurately for each $k$. Apart from $\xi^*$, we need to estimate $\nabla_\alpha C(\alpha, \theta)$ and the integral $\int_{\theta \in \Theta} \xi_\alpha^*(\theta)\mu_N(\theta)\nabla_\alpha C(\alpha, \theta)$.

To estimate $\nabla_\alpha C(\alpha, \theta)$, any plug-in estimation method satisfies our demand. Here, we adopt the variational policy gradient theorem inZhang et al. (2020), which consider the policy gradient for a concave function defined on the occupancy measure for a reinforcement learning problem. Different from our Bayesian-risk problem with a general loss function, Zhang et al. (2020) only considers the inner-layer $F$ of our objective equation 2 in the online setting. It should also be noticed that their method can be replaced by other methods such as the zeroth-order estimation method in Balasubramanian & Ghadimi (2022). While the variational policy gradient theorem require access to the conjugate function $F^*$, which may be difficult to calculate in some cases, zeroth-order method only requires function evaluation of $F$ but leads to higher computational cost in general cases.

**Lemma 3.1.** *(Theorem 3.1 in (Zhang et al., 2020)) Suppose $F$ is convex and continuously differentiable in an open neighborhood of $\lambda^{\pi_\alpha, P_\theta}$. Fix the transition kernel $P_\theta$ and denote $V(\alpha; z)$ to be the expected cumulative cost of policy $\pi_\alpha$ when the cost function is $z$, and assume $\nabla_\alpha V(\alpha; z)$ always exists. Then we have*

$$\nabla_\alpha C(\alpha, \theta) = -\lim_{\delta \to 0_+} \operatorname*{argmin}_{x \in \mathbb{R}^{SA}} \sup_{z \in \mathbb{R}^{SA}}\left\{V(\alpha; z) + \delta\nabla_\alpha V(\alpha; z)^\top x - F^*(z) + \frac{\delta}{2}\|x\|^2\right\}, \quad (8)$$

*where $V(\alpha; z) = \langle z, \lambda(\alpha, \theta)\rangle, \nabla_\alpha V(\alpha; z)^\top x = \langle z, \nabla_\alpha \lambda(\alpha, \theta)x\rangle, F^*(z) := \sup_{x \in \mathbb{R}^{SA}} x^\top z - F(x)$ is the Fenchel conjugate of $F$.*

A natural idea to calculate $\nabla_\alpha C$ is to use chain rule, i.e. $\nabla_\alpha C = \nabla_\lambda F \cdot \nabla_\alpha \lambda$. However, it may have a high computational cost if we directly estimate each part at a specific $\alpha$. The variational policy gradient method bypasses this issue by changing this problem into a problem of calculating some linear functions and the conjugate function at any $z$, shown in equation 8. To solve equation 8, we need to estimate $V(\alpha; z)$ and $\nabla_\alpha V(\alpha; z)$. Zhang et al. (2020) considers an online setting and thus they need to interact with the environment to estimate $\nabla_\alpha C$. In our offline setting, we can directly solve equation 8 to get $\nabla_\alpha C$. An example algorithm to solve equation 8 is given in Appendix B.2.

To evaluate the integral $\int_{\theta \in \Theta} \xi_\alpha^*(\theta)\mu_N(\theta)\nabla_\alpha C(\alpha, \theta)d\theta$, we use samples $\theta_k$ to construct the policy gradient estimator

$$\nabla_\alpha \rho_{\theta \sim \mu_N}(C(\alpha, \theta)) \approx \widehat{g}(\alpha) := \frac{1}{r_N}\sum_{k=1}^{r_N}\xi^*(\theta_k)\nabla_\alpha C(\alpha, \theta_k). \quad (9)$$

In this paper, we assume that we have access to samples from the posterior distribution $\mu_N$. In general, expensive methods such as Markov Chain Monte Carlo (MCMC) are often required to compute the posterior. However, by utilizing a conjugate prior, we obtain a closed-form expression for the posterior parameters, making the calculation more efficient. Bayesian update can also be done by neural network by normalizing the output of neural network into a probability. It should also be noted that we resort to solving the SAA problem equation 7 only when we cannot derive the closed-form expression for $\xi^*$, which depends on the risk measure we choose.

### 3.4 FULL ALGORITHM

To carry out policy gradient optimization, we iteratively use the following gradient descent step:

$$\alpha_{t+1} = \arg\min_{\alpha \in W} \langle \widehat{g}(\alpha_t), \alpha - \alpha_t \rangle + \frac{\eta_t}{2} \|\alpha - \alpha_t\|^2 = \text{Proj}_W \left( \alpha_t - \frac{1}{\eta_t} \widehat{g}(\alpha_t) \right), \qquad (10)$$

where $\eta_t$ is the step size, and $\text{Proj}_W(x) = \arg\min_{y \in W} \|y - x\|_2^2$ projects $x$ into the parameter space $W$. We summarize the full algorithm in Algorithm 1.

---

**Algorithm 1** Bayesian Risk Policy Gradient (BR-PG)

---

**input**: Initial $\alpha_0$, data $\zeta^{(N)}$ of size $N$, prior distribution $\mu_0(\theta)$, iteration number $T$;

Calculate the posterior $\mu_N(\theta) = \frac{P_\theta(\zeta^{(N)})\mu_0(\theta)}{\int_{\theta'} P_{\theta'}(\zeta^{(N)})\mu_0(\theta')}$;

**for** $t = 0$ to $T - 1$ **do**

Sample $\{\theta_k^t\}_{k=1}^{r_N}$ from $\mu_N(\theta)$;

Use the closed-form expression or solve the SAA problem equation 7 to get $\xi^*(\theta_k^t)$;

Solve the problem equation 8 to get $\nabla_\alpha C(\alpha_t, \theta_k^t)$ for $k = 1\ldots, r_N$;

$\widehat{g}_t := \frac{1}{r_N} \sum_{k=1}^{r_N} \xi^*(\theta_k^t) \nabla_\alpha C(\alpha, \theta_k^t)$;

$\alpha_{t+1} = \text{Proj}_W \left( \alpha_t - \frac{1}{\eta_t} \widehat{g}_t \right)$;

**end for**

**output**: $\alpha_T$.

---

### 3.5 EPISODIC SETTING

So far we have considered the offline setting with a fixed batch of data, but in many application problems data can be collected periodically. Again, consider a self-driving car as an example: the car is trained in an offline setting and then deployed to a real environment for a test drive while collecting more data from the environment. The collected data can be then used to learn about the environment and update the policy. This process can be repeated iteratively. Thus, we extend our approach to an episodic setting as described above. A potential approach is to use Algorithm 1 to make policy updates during each episode, as detailed in Algorithm 2 in Appendix A.2.

## 4 CONVERGENCE ANALYSIS

In this section, we analyze the convergence properties of Algorithm 1 and Algorithm 2. We begin by establishing the error bound for the policy gradient estimator. Next, we demonstrate the finite-time convergence rate is $\mathcal{O}((1 - \epsilon)^t)$, where $t$ represents the number of policy gradient iterations and $\epsilon$ is the accuracy. Furthermore, we prove the consistency of the proposed Bayesian risk formulation, meaning that the optimal policy obtained through this formulation converges to the one obtained by solving the true problem as the number of initial data points $N$ approaches infinity. Lastly, for the episodic setting we show the number of iterations required in any episode to maintain an $\mathcal{O}(\epsilon)$ optimality gap over all episodes.

### 4.1 ESTIMATION ERROR OF THE POLICY GRADIENT

**Assumption 4.1.** *Assume $\xi^*$ in Theorem 2 satisfies $\sup_{\alpha \in W} \text{Var}_{\theta \sim \mu_N}[\xi^*(\theta)\nabla C(\alpha, \theta)] = \sigma_\xi < \infty$.*

Assumption 4.1 requires $\xi^* \nabla C$ to have uniformly bounded variance. It is hard to show some property of $\xi^*$ in a general case as the envelope set is given in a general form. One sufficient condition for Assumption 4.1 to hold is that $\xi^*$ is bounded on $\Theta$. As $\Theta$ is a compact and convex set, it is not a strong condition.

**Theorem 3.** *Assume Assumption 3.1, 3.2 and 4.1 hold, and $\rho$ satisfies Definition 3.1. Then the gradient estimation error is*

$$\mathbb{E}\left[\|\widehat{g}(\alpha) - g(\alpha)\|_2^2\right] \leq \frac{\sigma_\xi}{r_N}, \forall \alpha \in W, \tag{11}$$

*where $r_N$ is the sample number for gradient estimator in equation 9.*

Theorem 3 implies that the sample complexity of $\Theta(1/\epsilon)$ is required to achieve the estimation error $\mathcal{O}(\epsilon)$. Please refer to Appendix B.3 for the detailed proof.

### 4.2 CONVERGENCE OF ALGORITHM 1

First we make an assumption about the Lipschitz continuity of $g(\alpha)$ in Assumption B.1. It should also be noted that although $\rho \circ F(\lambda)$ is convex w.r.t $\lambda$, the inner map $\lambda(\alpha)$ from policy parameter to occupancy measure is not necessarily convex in $\alpha$. However, the hidden convexity can be utilized to get the global optimality, regardless of the gradient estimation method. By utilizing the bijection assumption of $\lambda(\alpha)$, a first-order stationary point is still globally optimal, shown by Theorem 5.13 in Zhang et al. (2021), which requires the Assumption B.2. Assumption B.2 can be satisfied when $W$ is a compact convex set and $\lambda$ is a locally differentiable bijection.

**Theorem 4.** *(Optimality gap) Suppose that Assumption 3.1, 3.2, 4.1, B.1 and B.2 hold, and $\rho$ satisfies Definition 3.1. $\forall \epsilon < \bar{\epsilon}$ with $\bar{\epsilon}$ defined in Assumption B.2. By choosing $\eta_t = 2L_G$ in Algorithm 1, it holds that $\mathbb{E}G(\alpha_T) - G^* \leq (1 - \epsilon)^T \left[\mathbb{E}G(\alpha_0) - G^*\right] + \mathcal{O}(\epsilon + r_N^{-1}\epsilon^{-1})$, where $G^*$ is the globally minimal value of $G$.*

Theorem 4 shows the optimality gap of the objective value consisting of two parts: an asymptotically diminishing error bound with factor $(1-\epsilon)^T$ in the exact setting and an estimation error bound of the policy gradient. The samples are from the posterior $\mu_N$ and the total sample complexity to achieve accuracy $\mathcal{O}(\epsilon)$ is $\mathcal{O}(\epsilon^{-3}\log(\epsilon^{-1}))$ by choosing $T = \log_2(\frac{\mathbb{E}G(\alpha_0) - G(\alpha^*)}{\epsilon})\epsilon^{-1}$ and $r_N = \epsilon^{-2}$. The proof and assumptions are shown in Appendix B.4.

**Theorem 5.** *(Consistency) Suppose that Assumption 3.1, 3.2, 4.1, B.1, B.2 and B.3 hold, and $\rho$ satisfies Definition 3.1. Then we have $\sup_\alpha |\rho_{\theta \sim \mu_i}(C(\alpha, \theta)) - C(\alpha, \theta^*)|$ tends to 0 with probability 1 as $i \to \infty$, where the probability is w.r.t. infinite product probability measure of the data sequence. Moreover, $C(\alpha_N^*, \theta^*) - C(\alpha^*, \theta^*) \to 0$ with probability 1 as $N \to \infty$, where $\alpha_N^*$ is a global minimizer of $\rho_{\theta \sim \mu_N}(C(\alpha, \theta))$ and $\alpha^*$ is a global minimizer of $C(\alpha, \theta^*)$.*

As the data size $N$ increases, the posterior distribution converges to a Dirac measure, which is a point mass at the true parameter $\theta^*$. Consequently, the performance of the optimal policy for the posterior $\mu_N$ converges to the optimal policy under the true environment, as demonstrated in Theorem 5. Since we consider a series of posteriors rather than a fixed posterior, as discussed earlier, additional assumptions are required to ensure the convergence of the posteriors. Broadly speaking, it is necessary that all parameters and all data points have positive probabilities of being sampled under both the prior and posterior distributions, and that the interchangeability of limits and integrals is satisfied. Detailed proof and assumptions are provided in Appendix B.5.

In the episodic setting, we iteratively use the current policy for data collection and posterior updates, and perform policy updates based on the updated posterior, as described in Algorithm 2. A natural question arises: how many iterations are required within a given episode to achieve a certain level of accuracy? This is addressed in Theorem 6.

**Theorem 6.** *Suppose that Assumption 3.1, 3.2, 4.1, B.1, B.2 and B.3 hold, and $\rho$ satisfies Definition 3.1. Assume that $G_i(\alpha) := \rho_{\theta \sim \mu_i}(C(\alpha, \theta))$, which is the objective for the $i$-th episode, has $L_{G,i}$-Lipschitz continuous gradient. Let $\{\alpha_{i,j}\}, i = 1, \ldots, N, j = 0, \ldots, t_i$ be the generated policy parameter sequences for $N$ episodes by Algorithm 2, where $\alpha_{i+1,0} = \alpha_{i,t_i}$. For any $0 < \epsilon < \bar{\epsilon}$ with $\bar{\epsilon}$ defined in Assumption B.2, if we want to keep a constant error bound $\mathcal{O}(\epsilon)$ for $\mathbb{E}[G_i(\alpha_{i,t_i}) - G_i(\alpha_i^*)], i = 1, \cdots, N$, then we need the sample number to be $r_i = \Theta(\epsilon^{-2}/L_{G,i})$ and $t_i$ to be at*

most $\mathcal{O}(\epsilon^{-1}\log(\frac{D_i+D_{i-1}}{\epsilon}))$, where $D_i := \sup_\alpha |\rho_{\theta \sim \mu_i}(C(\alpha, \theta)) - C(\alpha, \theta^*)|$ converges to $0$ when $i \to \infty$ as shown in Theorem 5.

Theorem 6 offers theoretical advice on how to choose the iteration number in each episode. When $i$ is small, we choose $t_i$ to be $\Theta(\epsilon^{-1}\log(\epsilon^{-1}))$. When $i$ is large, we do not need as many iterations to keep the optimality gap since $D_i$ approaches $0$. Detailed proof can be found in Appendix B.6.

## 5 NUMERICAL EXPERIMENTS

We demonstrate the performance of our proposed formulation and algorithm using an offline planning problem known as the frozen lake problem (Ravichandiran, 2018), an Open AI benchmark. For a detailed description of the problem, we refer readers to Appendix F. We consider different convex loss functions, including the mean and Kullback-Leibler (KL) divergence, for various tasks.

Table 1: Results for frozen lake problem. Linear loss and positive-sided variance at different risk levels $\alpha$ are reported for different algorithms and different data sizes with linear loss function. Standard errors are reported in parentheses. Escape probability $\theta_e = 0.02$ and number of data points is $N = 5$ and $50$.

| Approach | N=5 | | N=50 | |
|---|---|---|---|---|
| | linear loss | positive-sided variance | linear loss | positive-sided variance |
| BR-PG ($\beta = 0$) | 33.886(0.347 ) | 5.212 | 32.784 (0.00825) | 0.0026 |
| BR-PG ($\beta = 0.5$) | 33.104 (0.127) | 0.710 | 32.757 (0.00516) | 0.00119 |
| BR-PG ($\beta = 0.9$) | 32.854 (0.0641) | 0.193 | 32.741 (0.00283) | 0.000376 |
| Empirical Method | 37.057(0.927) | 34.387 | 33.340 (0.0936) | 0.380 |
| DRQL(radius=0.05) | 37.936(0.887) | 26.554 | 34.365(0.366) | 5.139 |
| DRQL(radius=1) | 35.216(0.732) | 22.213 | 32.924(0.105) | 0.519 |
| DRQL(radius=20) | 36.255(0.813) | 24.622 | 32.855(0.063) | 0.179 |
| Optimal Policy under True Model | 32.499 | | 32.499 | |

We compare the Bayesian Risk Policy Gradient (BR-PG) algorithm with CVaR risk measure under different risk levels $\beta = 0, 0.5, 0.9$, respectively, with two other methods. The first is the benchmark empirical approach, which computes a maximum likelihood estimator (MLE) for the parameters from the given dataset and obtains a policy by solving the MDP with the plugged-in MLE parameter value. The second method is an offline version of distributionally robust Q-learning (DRQL) algorithm (Liu et al., 2022), which uses Q-learning to find the best policy in the worst-case distributional perturbation of the environment. (Liu et al., 2022) adopt a KL divergence ball centered at the true transition kernel as the environment's perturbation. When the risk level $\beta$ approaches 1, Bayesian-risk performance is similar as the worst-case performance. Since we are considering an offline planning problem, we modify the DRQL to interact with an offline simulator that uses the transition kernel with the MLE parameters derived from the data. In other words, DRQL minimizes the worst-case performance for a KL divergence ball centered at the MLE kernel. For a fair comparison, we conduct DRQL experiments with different radii of the KL divergence ball.

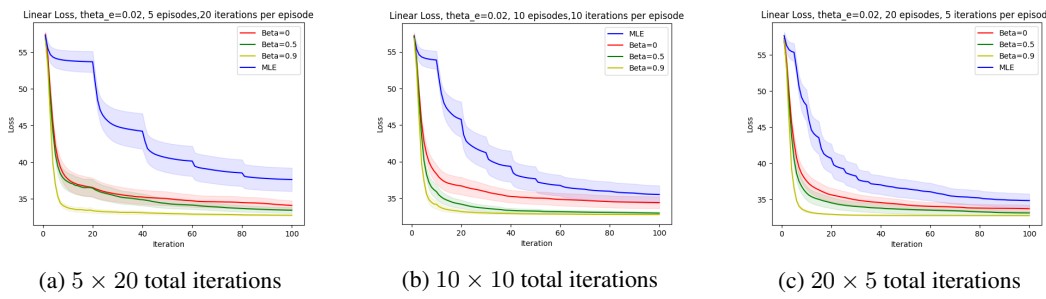

(a) $5 \times 20$ total iterations  (b) $10 \times 10$ total iterations  (c) $20 \times 5$ total iterations

Figure 1: Results for episodic case with different episode numbers and iterations per episode under the same escape probability $\theta_e = 0.02$ and 50 replications. Here the loss function is still chosen to be the linear loss. 95% confidence intervals are reported by the shaded bands.

**Linear Loss.** We consider the linear loss function, which corresponds to the total discounted cost in a classical MDP problem. This is referred to as one replication, and we repeat for 50 replications using different independent data sets. More details can be found in Appendix F.

**Episodic Case**. We consider the episodic setting where the data collection and policy update are alternatively conducted. More implement details can be found in Appendix F for 50 replications.

**Mimicking a policy**. Here we consider a different problem of mimicking an expert policy still using Frozen Lake environment and 50 replications. The loss function we want to minimize is defined as the KL divergence between state occupancy measure under the current policy and the expert state distribution. More implement details can be found in Appendix F.

**Conclusions.** In each replication, data points are randomly sampled from the true distribution. While facing the epistemic uncertainty, BR-PG algorithm provides robustness across different loss functions. Table 1 shows that our proposed BR-PG algorithm has lower linear loss, standard error and positive-sided variance (psv), demonstrating more robustness in the sense of balancing the mean and variability of the actual cost. In contrast, the empirical approach performs badly when the data size is small, e.g. $N = 5$, indicating that it is not robust against the epistemic uncertainty and suffers from the scarcity of data. DRQL also performs better than empirical method but worse than our algorithm in the sense of having larger mean and variance of the loss. Figure 1 shows that the loss of our algorithm decreases quickly in spite of few data. In the episodic case, the loss function decreases faster with more episodes (but the same total number of iterations), due to more collected data with more episodes. The loss function of our BR-PG method decreases more quickly in early episodes, which is shown by two differences between Figure 2a and Figure 2b. First, the 95% confidence interval, shown in the shaded band around each curve, is narrower for $N = 50$. Second, the absolute loss of $N = 50$ decreases by about 20% compared with $N = 5$. Figure 2 demonstrates the better performance of our proposed BR-PG algorithm compared to the empirical approach, where we achieve smaller loss and lower variability, for the policy mimicking task. From Table 1 and Figure 1, we can see when there are more data, the posterior distribution used in BR-PG algorithm and the MLE estimator used in the empirical approach converges to the true parameter as data size increases, which reduces to solving an MDP with known transition probability, and therefore, the optimal policies and the actual costs tend to be similar.

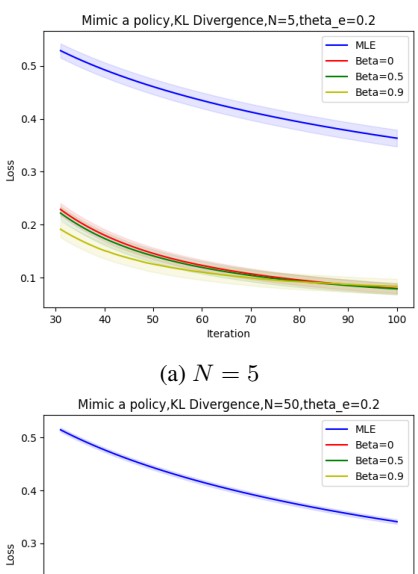

(a) $N = 5$

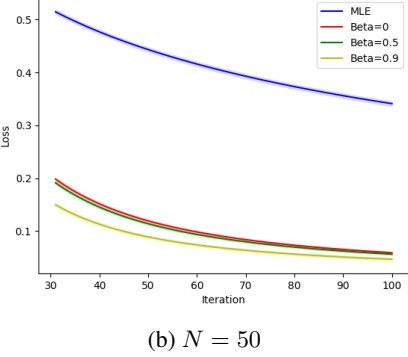

(b) $N = 50$

Figure 2: Results for loss function "KL Divergence" with data sizes $N = 5$ and 50 under $\theta_e = 0.02$. 95% confidence intervals are reported in the shaded area.

## 6 CONCLUSIONS

In this paper, we develop a Bayesian risk approach to jointly address both epistemic and intrinsic uncertainty in the infinite-horizon MDP. For a general coherent risk measure and a general convex loss function, we design a policy gradient algorithm for the proposed formulation and demonstrate the algorithm's convergence at a rate of $\mathcal{O}((1 - \epsilon)^t)$. Furthermore, we establish the consistency of an online episodic extension and provide bounds on the number of iterations required to maintain an error bound $\mathcal{O}(\epsilon)$ for each episode. The numerical experiments confirm the convergence of the proposed algorithm and demonstrate the robustness of the formulation under various loss functions.

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

# A APPENDIX

## A.1 REVIEW ON CONVEX RL

Our problem is highly relevant to convex RL, which generalizes cumulative reward on a convex general-utility objective instead of cumulative reward. Specifically, our problem is closely tied to convex RL, which extends the traditional cumulative reward framework to a convex general-utility objective. Prior research has explored policy gradient methods to address convex RL. For instance, Zhang et al. (2020) demonstrates that the policy gradient of convex RL can be formulated as a min-max optimization problem. To reduce estimator variance, Zhang et al. (2021) introduces an off-policy policy gradient estimator that leverages mini-batch techniques and truncation mechanisms, while Barakat et al. (2023) employs a recursive approach to handle large state-action spaces. In the domain of multi-agent convex RL, Zhang et al. (2022) assumes global state observability and proposes a trajectory-based actor-critic method. Recent studies have also focused on safe convex RL, where the objective is to maximize a convex utility function under convex safety constraints. For example, Ying et al. (2023) develops a primal-dual algorithm with strong guarantees on the optimality gap and constraint violations, achieving an $\mathcal{O}(1/\epsilon^3)$ bound in the convex-concave case with zero constraint violation. Building on this, Bai et al. (2023) improves the bound to $\mathcal{O}(1/\epsilon^2)$. Furthermore, Ying et al. (2024) extends the primal-dual framework to multi-agent convex safe RL.

## A.2 ALGORITHMS

---

**Algorithm 2** Episodic BR-PG

---

**input**: Initial $\alpha_0$, prior distribution $\mu_0(\theta)$, total episode number $N$.
Deploy policy $\pi(\alpha_0)$ to gain the initial data set $\zeta^{(1)}$.
**for** $i = 1$ to $N$ **do**
    Let $\alpha_{i,0} := \alpha_{i-1,t_{i-1}}$, where $\alpha_{0,t_0} := \alpha_0$;
    Calculate the posterior $\mu_i(\theta) = \frac{P_\theta(\zeta^{(i)})\mu_{i-1}(\theta)}{\int_{\theta'} P_{\theta'}(\zeta^{(i)})\mu_{i-1}(\theta')}$;
    Use Algorithm 1 with $t_i$ iterations and initial point $\alpha_{i,0}$ to generate the policy parameter sequence $\alpha_{i,1}, \cdots, \alpha_{i,t_i}$.
    **if** $i < N$ **then**
        Deploy policy $\pi(\alpha_{i,t_i})$ and gain a new data set $\zeta^{(i+1)}$.
    **end if**
**end for**
**output**: $\alpha_T$.

---

# B PROOF DETAILS

## B.1 PROOF OF THEOREM 2

*Proof.*

$$\mathcal{U}(\mu_N) = \{\xi : g_e(\xi, \mu_N) = 0, \forall e \in \mathcal{E}, f_i(\xi, \mu_N) \leq 0, \forall i \in \mathcal{I},$$
$$\int_{\theta \in \Theta} \xi(\theta)\mu_N(\theta) = 1, \xi(\theta) \geq 0, \|\xi\|_q \leq B_q\}.$$

Define the Lagrangian:

$$L_\alpha(\xi, \lambda^\mathcal{I}, \lambda^\mathcal{E}) = \int_{\theta \in \Theta} \xi(\theta)\mu_N(\theta)C(\alpha, \theta) - \sum_{i \in \mathcal{I}} \lambda^\mathcal{I}(i)f_i(\xi, \mu_N) - \sum_{e \in \mathcal{E}} \lambda^\mathcal{E}(e)g_e(\xi, \mu_N), \quad (12)$$

and a subtly relaxed envelope

$$\mathcal{U}'(\mu_N) = \{\xi : \int_{\theta \in \Theta} \xi(\theta)\mu_N(\theta) = 1, \xi(\theta) \geq 0, , \|\xi\|_q \leq B_q\}.$$

As mentioned before, we can rewrite the objective as the value of a max-min problem in equation 4

$$\rho_{\theta \sim \mu_N}(C(\alpha, \theta)) = \max_{\xi \in \mathcal{U}'(\mu_N)} \min_{\lambda^{\mathcal{I}} \geq 0, \lambda^{\mathcal{E}}} L_\alpha(\xi, \lambda^{\mathcal{I}}, \lambda^{\mathcal{E}}).$$

Two things deserved to be noticed: (i) Slater's condition holds in the primal optimization problem equation 4 by Definition 3.1. (ii) $L_\theta\left(\xi, \lambda^{\mathcal{I}}, \lambda^{\mathcal{E}}\right)$ is concave in $\xi$ and convex in $(\lambda^{\mathcal{I}}, \lambda^{\mathcal{E}})$. Then strong duality holds for equation 4.

$$\begin{aligned}
\rho_{\theta \sim \mu_N}(C(\alpha, \theta)) &= \max_{\xi \in \mathcal{U}'(\mu_N)} \min_{\lambda^{\mathcal{I}} \geq 0, \lambda^{\mathcal{E}}} L_\alpha(\xi, \lambda^{\mathcal{I}}, \lambda^{\mathcal{E}}) \\
&= \min_{\lambda^{\mathcal{I}} \geq 0, \lambda^{\mathcal{E}}} \max_{\xi \in \mathcal{U}'(\mu_N)} L_\alpha(\xi, \lambda^{\mathcal{I}}, \lambda^{\mathcal{E}})
\end{aligned} \tag{13}$$

As $\nabla_\alpha C(\alpha, \theta)$ is uniformly bounded for all $\theta$ and $\alpha$, we have $\nabla_\alpha L_\alpha(\xi, \lambda^{\mathcal{I}}, \lambda^{\mathcal{E}})$ is uniformly bounded w.r.t $\alpha$ and continuous at all$(\xi, \lambda^{\mathcal{I}}, \lambda^{\mathcal{E}})$. Then we have $L_\alpha(\xi, \lambda^{\mathcal{I}}, \lambda^{\mathcal{E}})$ is absolutely continuous w.r.t $\alpha$ for all $(\xi, \lambda^{\mathcal{I}}, \lambda^{\mathcal{E}})$. Since $\nabla_\alpha^2 C(\alpha, \theta)$ is uniformly bounded for all $\theta$ and $\alpha$, we have $\{L_\alpha(\xi, \lambda^{\mathcal{I}}, \lambda^{\mathcal{E}})\}_{(\xi, \lambda^{\mathcal{I}}, \lambda^{\mathcal{E}})}$ is equi-differentiable in $\alpha$.

As $\Theta$ is compact and convex, $\Theta$ is a separable metric space with Euclidean metric and its Borel sigma algebra. Then $(\Theta, \mu_N)$ is a separable metric measure space. By Theorem 4.13 (Brezis & Brézis, 2011), $L^q(\Theta, \mu_N)$ is separable. Then $\mathcal{U}'(\mu_N) = \{\xi \in L^q(\Theta, \mu_N) : \int_{\theta \in \Theta} \xi(\theta)\mu_N(\theta) = 1, \xi(\theta) \geq 0,, \|\xi\|_q \leq B_q\}d$ is separable.

Define the set of saddle point for equation 13 by $X^* = \arg\max_{\xi \in \mathcal{U}'(\mu_N)} \min_{\lambda^{\mathcal{I}} \geq 0, \lambda^{\mathcal{E}}} L_\alpha(\xi, \lambda^{\mathcal{I}}, \lambda^{\mathcal{E}})$ and $Y^* = \arg\min_{\lambda^{\mathcal{I}} \geq 0, \lambda^{\mathcal{E}}} \max_{\xi \in \mathcal{U}'(\mu_N)} L_\alpha(\xi, \lambda^{\mathcal{I}}, \lambda^{\mathcal{E}})$.

Then for every selection of saddle point $\left(\xi_\alpha^*, \lambda_\alpha^{*,\mathcal{E}}, \lambda_\alpha^{*,\mathcal{I}}\right) \in X^* \times Y^*$, the Envelope theorem for saddle-point problems ( Theorem 4(Milgrom & Segal, 2002) ) shows that

$$\begin{aligned}
\nabla_\alpha \rho_{\theta \sim \mu_N}(C(\alpha, \theta)) &= \nabla_\alpha \max_{\xi \in \mathcal{U}'(\mu_N)} \min_{\lambda^{\mathcal{I}} \geq 0, \lambda^{\mathcal{E}}} L_\alpha(\xi, \lambda^{\mathcal{I}}, \lambda^{\mathcal{E}}) \\
&= \nabla_\alpha L_\alpha(\xi, \lambda^{\mathcal{I}}, \lambda^{\mathcal{E}})\Big|_{\left(\xi_\alpha^*, \lambda_\alpha^{*,\mathcal{E}}, \lambda_\alpha^{*,\mathcal{I}}\right)} \\
&= \int_{\theta \in \Theta} \xi_\alpha^*(\theta)\mu_N(\theta)\nabla_\alpha C(\alpha, \theta)
\end{aligned} \tag{14}$$

$\square$

## B.2 PROOF OF LEMMA 3.1

*Proof.* Here is a brief proof sketch, and the full proof can be found in the proof of Theorem 3.1(Zhang et al., 2020). For a convex function, the conjugate of the conjugate is itself. Notice that $V(\alpha; z) + \delta \nabla_\alpha V(\alpha; z)^\top x - F^*(z) = \langle z, \lambda(\alpha, \theta) + \delta \nabla_\alpha \lambda(\alpha, \theta)x \rangle - F^*(z)$. Then we have $\sup_{z \in \mathbb{R}^{SA}} V(\alpha; z) + \delta \nabla_\alpha V(\alpha; z)^\top x - F^*(z) = F(\lambda(\alpha, \theta) + \delta \nabla_\alpha \lambda(\alpha, \theta)x)$. By the first-order condition, we have

$$\operatorname*{argmin}_{x \in \mathbb{R}^{SA}} F(\lambda(\alpha, \theta) + \delta \nabla_\alpha \lambda(\alpha, \theta)x) + \frac{\delta}{2}\|x\|_2^2 = -\nabla F\left(\lambda(\alpha, \theta) + \delta \nabla_\alpha \lambda(\alpha, \theta)x\right) \nabla_\alpha \lambda(\alpha, \theta)x.$$

By letting $\delta \to 0^+$ and using the chain rule, we get the result equation 8. $\square$

### B.2.1 ALGORITHM FOR SOLVING THEOREM 3.1

**Estimate** $V(\alpha, z)$**:** Recall that we consider an offline setting where the transition kernel $P_\theta$ is assumed to be known for any given $\theta$. For any fixed transition kernel $P_\theta$ and policy $\pi_\alpha$, we can estimate the occupancy measure by making a truncation $K$ in the definition of occupancy measure in equation 1:

$$\widehat{\lambda_{sa}^{\pi,P}} = \sum_{t=0}^{K} \gamma^t \cdot \mathbb{P}\left(s_t = s, a_t = a \mid \pi, s_0 \sim \tau, P\right)$$

with the error $\|\widehat{\lambda} - \lambda\|_1 \leq \epsilon_\lambda := \gamma^K/(1-\gamma)$. This error can be made arbitrarily small by increasing $K$, thus we assume that we can exactly compute occupancy measure. After computing the occupancy measure, $V(\alpha; z) = \langle z, \lambda \rangle$.

**Estimate** $\nabla_\alpha V(\alpha, z)$**:** The policy gradient theorem (Sutton et al., 1999) shows that

$$\nabla_\alpha V(\alpha; z) = \mathbb{E}^{\pi_\alpha} \left[ \sum_{t=0}^\infty \gamma^t Q^{\pi_\alpha}(s_t, a_t; z) \cdot \nabla_\alpha \log \pi_\alpha(a_t \mid s_t) \right]$$

where $Q^\pi(s, a; z) := \mathbb{E}^\pi \left[ \sum_{t=0}^\infty \gamma^t z(s_t, a_t) \mid s_0 = s, a_0 = a, a_t \sim \pi(\cdot \mid s_t) \right]$ satisfying the Bellman equation

$$Q^\pi(s, a; z) = \mathbb{E}[z(s, a)] + \sum_{s' \in \mathcal{S}} \sum_{a' \in \mathcal{A}} P_\theta(s'|s, a)\pi(a'|s')Q^\pi(s', a'; z). \tag{15}$$

For any given $\theta$, policy $\pi$ and cost function $z$, we can solve the Bellman equation equation 15 exactly to get $Q(\cdot, \cdot)$. It can be seen that $\nabla_\alpha V(\alpha; z)$ is a linear function of $\lambda$:

$$\nabla_\alpha V(\alpha; z) = \sum_{s \in \mathcal{S}} \sum_{a \in \mathcal{A}} Q(s, a) \dot{\nabla}_\alpha \log \pi_\alpha(a \mid s) \lambda(s, a).$$

For any $\theta$, policy $\pi$ and cost function $z$, we can calculate the $Q$ value function by solving the Bellman equation:

$$Q^\pi(s, a; z) = \mathbb{E}[z(s, a)] + \sum_{s' \in \mathcal{S}} \sum_{a' \in \mathcal{A}} P_\theta(s'|s, a)\pi(a'|s')Q^\pi(s', a'; z)$$

Then we can use Algorithm 3 to solve equation 8 in Lemma 3.1. It should be noticed that $\delta \nabla_\alpha V(\alpha; z)^\top x = \mathcal{O}(\delta)$ is omitted when calculating the gradient for $z$ as $\delta \to 0$. Thus we omit this term when calculating the gradient for $z$. To evaluate the integral $\int_{\theta \in \Theta} \xi_\alpha^*(\theta)\mu_N(\theta)\nabla_\alpha C(\alpha, \theta)$, we sample i.i.d $\theta_k$ from $\mu_N$ for $k = 1, \ldots, r_N$, then we can construct the policy gradient estimator

$$\nabla_\alpha \rho_{\theta \sim \mu_N}(C(\alpha, \theta)) \approx \widehat{g}(\alpha) := \frac{1}{r_N} \sum_{k=1}^{r_N} \xi^*(\theta_k)\nabla_\alpha C(\alpha, \theta_k).$$

---

**Algorithm 3** Alternative Gradient Descent Method

**input**: initial $z_0, x_0$, step sizes $a_t, b_t$, iteration number $T$, transition kernel parameter $\theta$, policy parameter $\alpha$;
**for** $t = 0$ to $T - 1$ **do**
$\quad z_{t+1} = z_t + a_t[\lambda(\alpha, \theta) - \nabla F^*(z_t)]$
$\quad x_{t+1} = x_t - b_t[\nabla_\alpha V(\alpha; z) + x_t]$, where $\nabla_\alpha V(\alpha; z) = \sum_{s,a} Q(s, a)\dot{\nabla}_\alpha \log \pi_\alpha(a \mid s) \lambda(s, a)$
**end for**
**output**: $-x_T$.

---

### B.3 PROOF OF THEOREM 3

*Proof.* By Theorem 2, the true gradient is

$$g(\alpha) = \int_{\theta \in \Theta} \xi_\alpha^*(\theta)\mu_N(\theta)\nabla_\alpha C(\alpha, \theta).$$

And our gradient estimator is

$$\widehat{g}(\alpha) := \frac{1}{r_N} \sum_{k=1}^{r_N} \xi^*(\theta_k)\nabla_\alpha C(\alpha, \theta_k).$$

Then we have

$$\mathbb{E}\|\widehat{g} - g\|_2^2 \leq \frac{1}{r_N}\mathbb{E}\|\xi^*(\theta_1)\nabla_\alpha C(\alpha, \theta_1) - \int_\Theta \xi^*(\theta)\mu_N(\theta)\nabla_\alpha C(\alpha, \theta)d\theta\|_2^2 \leq \frac{\sigma_\xi}{r_N}.$$

$\square$

### B.4 PROOF OF THEOREM 4

First, we make an assumption about $G$.

**Assumption B.1.** *There exists some $L_G > 0$ s.t. $g(\alpha)$ is $L_G$-Lipschitz continuous in $\alpha$.*

Then we need assumptions about the mapping from policy parameter to occupancy.

**Assumption B.2.** *(Assumption 5.11 in Zhang et al. (2021)) For policy parameterization $\pi_\alpha$, $\alpha$ overparametrizes the set of policies in the following sense. (i). For any $\alpha$ and $\lambda(\alpha)$, there exist (relative) neighourhoods $\alpha \in \mathcal{U}_\alpha \subset W$ and $\lambda(\alpha) \in \mathcal{V}_{\lambda(\alpha)} \subset \lambda(W)$ s.t. $\left(\lambda\,\middle|\,_{\mathcal{U}_\alpha}\right)(\cdot)$ forms a bijection between $\mathcal{U}_\alpha$ and $\mathcal{V}_{\lambda(\alpha)}$, where $\left(\lambda\,|\,\mathcal{U}_\alpha\right)(\cdot)$ is the confinement of $\lambda$ onto $\mathcal{U}_\alpha$. We assume $\left(\lambda\,|\,\mathcal{U}_\alpha\right)^{-1}(\cdot)$ is $\ell_\alpha$-Lipschitz continuous for any $\alpha$. (ii). Let $\pi_{\alpha^*}$ be the optimal policy. Assume there exists $\bar\epsilon$ small enough, s.t. $(1-\epsilon)\lambda(\alpha) + \epsilon\lambda(\alpha^*) \in \mathcal{V}_{\lambda(\alpha)}$ for $\forall \epsilon \leq \bar\epsilon, \forall\alpha$.*

*Proof.* For ease of notation, denote $g(\alpha_t)$ as $g_t$ and $\widehat{g}(\alpha_t)$ as $\widehat{g}_t$. By Assumption B.1, we have

$$
\begin{aligned}
G(\alpha) &\leq G(\alpha_t) + \langle g_t, \alpha - \alpha_t \rangle + \frac{L_G}{2}\|\alpha - \alpha_t\|_2^2 \\
&\leq G(\alpha) + L_G\|\alpha - \alpha_t\|_2^2.
\end{aligned}
\tag{16}
$$

Then we have

$$
\begin{aligned}
G(\alpha_{t+1}) &\leq G(\alpha_t) + \langle \widehat{g}_t, \alpha_{t+1} - \alpha_t \rangle + \langle g_t - \widehat{g}_t, \alpha_{t+1} - \alpha_t \rangle + \frac{L_G}{2}\|\alpha_{t+1} - \alpha_t\|_2^2 \\
&\leq G(\alpha_t) + \langle \widehat{g}_t, \alpha_{t+1} - \alpha_t \rangle + \frac{1}{2L_G}\|g_t - \widehat{g}_t\|_2^2 + \frac{L_G}{2}\|\alpha_{t+1} - \alpha_t\|_2^2 + \frac{L_G}{2}\|\alpha_{t+1} - \alpha_t\|_2^2 \\
&= G(\alpha_t) + \langle \widehat{g}_t, \alpha_{t+1} - \alpha_t \rangle + \frac{1}{2L_G}\|g_t - \widehat{g}_t\|_2^2 + L_G\|\alpha_{t+1} - \alpha_t\|_2^2 \\
&= \min_{\alpha \in W} G(\alpha_t) + \langle \widehat{g}_t, \alpha - \alpha_t \rangle + L_G\|\alpha - \alpha_t\|_2^2 + \frac{1}{2L_G}\|g_t - \widehat{g}_t\|_2^2 \\
&= \min_{\alpha \in W} G(\alpha_t) + \langle g_t, \alpha - \alpha_t \rangle + L_G\|\alpha - \alpha_t\|_2^2 + \langle \widehat{g}_t - g_t, \alpha - \alpha_t \rangle + \frac{1}{2L_G}\|g_t - \widehat{g}_t\|_2^2 \\
&\leq \min_{\alpha \in W} G(\alpha) + \frac{3L_G}{2}\|\alpha - \alpha_t\|_2^2 + \frac{L_G}{2}\|\alpha - \alpha_t\|_2^2 + \frac{1}{2L_G}\|g_t - \widehat{g}_t\|_2^2 + \frac{1}{2L_G}\|g_t - \widehat{g}_t\|_2^2 \\
&= \min_{\alpha \in W} G(\alpha) + 2L_G\|\alpha - \alpha_t\|_2^2 + \frac{1}{L_G}\|g_t - \widehat{g}_t\|_2^2,
\end{aligned}
$$

where the first inequality comes from equation 16, the second inequality comes from Cauchy–Schwarz inequality, the second equality holds because the definition of $\alpha_{t+1}$, the third inequality holds because of equation 16 and Cauchy–Schwarz inequality again.

For any $\epsilon < \bar\epsilon$, by Assumption B.2, $(1-\epsilon)\lambda(\alpha_t) + \epsilon\lambda(\alpha^*) \in \mathcal{V}_{\lambda(\alpha_t)}$ and thus

$$
\alpha_\epsilon := \left(\lambda\,|\,\mathcal{U}_{\alpha_t}\right)^{-1}\left((1-\epsilon)\lambda(\alpha_t) + \epsilon\lambda(\alpha^*)\right) \in \mathcal{U}_{\alpha_t} \subset W.
\tag{17}
$$

Then

$$
G(\alpha_{t+1}) \leq G(\alpha_\epsilon) + 2L_G\|\alpha_\epsilon - \alpha_t\|_2^2 + \frac{1}{L_G}\|g_t - \widehat{g}_t\|_2^2
\tag{18}
$$

Notice that

$$
G(\alpha_\epsilon) = F((1-\epsilon)\lambda(\alpha_t) + \epsilon\lambda(\alpha^*)) \leq (1-\epsilon)G(\alpha_t) + \epsilon G(\alpha^*)
\tag{19}
$$

Also,

$$
\begin{aligned}
\|\alpha_\epsilon - \alpha_t\|_2^2 &= \|\left(\lambda\,|\,\mathcal{U}_{\alpha_t}\right)^{-1}\left((1-\epsilon)\lambda(\alpha_t) + \epsilon\lambda(\alpha^*)\right) - \left(\lambda\,|\,\mathcal{U}_{\alpha_t}\right)^{-1}\left(\lambda(\alpha_t)\right)\|_2^2 \\
&\leq \ell_\alpha \epsilon^2 \|\lambda(\alpha_t) - \lambda(\alpha^*)\|_2^2 \\
&\leq \ell_\alpha \epsilon^2 D_\lambda^2,
\end{aligned}
\tag{20}
$$

where $D_\lambda := \sup_{\lambda,\lambda' \in \lambda(W)} \|\lambda - \lambda'\|_2$

By Lemma 3, $\mathbb{E}[\|g_t - \widehat{g}_t\|_2^2] \leq \frac{\sigma_\xi}{r_N}$. Substitute all these things into equation 18, we have

$$\mathbb{E}G(\alpha_{t+1}) \leq (1-\epsilon)\mathbb{E}G(\alpha_t) + \epsilon G(\alpha^*) + 2L_G\ell_\alpha\epsilon^2 D_\lambda^2 + \frac{\sigma_\xi}{r_N L_G}.$$

Then it holds that

$$\mathbb{E}G(\alpha_{t+1}) - G(\alpha^*) \leq (1-\epsilon)\left[\mathbb{E}G(\alpha_t) - G(\alpha^*)\right] + 2L_G\ell_\alpha D_\lambda^2\epsilon^2 + \frac{\sigma_\xi}{r_N L_G}. \tag{21}$$

Telescoping equation 21 over $t$ shows that

$$\mathbb{E}G(\alpha_T) - G(\alpha^*) \leq (1-\epsilon)^T\left[\mathbb{E}G(\alpha_0) - G(\alpha^*)\right] + 2L_G\ell_\alpha D_\lambda^2\epsilon + \frac{\sigma_\xi}{r_N L_G \epsilon} \tag{22}$$

Note that $(1-\epsilon)^{\epsilon^{-1}} \leq 1/2, \forall \epsilon \leq 1$. Choosing $T = \log_2(\frac{\mathbb{E}G(\alpha_0) - G(\alpha^*)}{\epsilon})\epsilon^{-1}$ and $r_N = \epsilon^{-2}$, we have

$$\mathbb{E}G(\alpha_T) - G(\alpha^*) \leq (1 + 2L_G\ell_\alpha D_\lambda^2 + \frac{\sigma_\xi}{L_G})\epsilon.$$

$\square$

### B.5 PROOF OF THEOREM 5

**Assumption B.3.** *(Assumption 3.1 in (Shapiro et al., 2023))*

*(1) The set $\Theta$ is convex compact with nonempty interior.*

*(2) $\ln \mu_0(\theta)$ is bounded on $\Theta$, i.e., there are constants $c_1 > c_2 > 0$ such that $c_1 \geq \mu_0(\theta) \geq c_2$ for all $\theta \in \Theta$.*

*(3) $P^*(\zeta) > 0$ for any $\zeta \in \Xi$.*

*(4) $P_\theta(\zeta) > 0$, and hence $\mu_N(\theta) > 0$, for all $\xi \in \Xi$ and $\theta \in \Theta$.*

*(5) $P_\zeta(\xi)$ is continuous in $\theta \in \Theta$.*

*(6) $\ln P_\theta(\zeta), \theta \in \Theta$, is dominated by an integrable (w.r.t. $P_*$) function.*

Assumption B.3 (1), (2) are used to guarantee the uniform convergence of posterior. Assumption B.3 (3), (4) require that all data points has positive probability to be sampled under the prior and posterior. Assumption B.3 (5), (6) are used to exchange the order of limit and integral.

With Assumption B.3, we are now ready to prove Theorem 5. Define a function $\psi(\theta) = \mathbb{E}_{P^*}[\ln P_\theta(\xi)]$ and let $\Theta^* := \{\theta' \in \Theta : \psi(\theta') = \inf_{\theta \in \Theta} \psi(\theta)\}$. For $\epsilon > 0$, define sets

$$V_\epsilon := \{\theta \in \Theta : \psi(\theta^*) - \psi(\theta) \geq \epsilon\}, U_\epsilon := \Theta \backslash V_\epsilon = \{\theta \in \Theta : \psi(\theta^*) - \psi(\theta) < \epsilon\}.$$

First we need to show two intermediate lemmas.

**Lemma B.1.** *(Lemma 3.1. (Shapiro et al., 2023)) Suppose that Assumption B.3 holds. Then for $0 < \epsilon_2 < \epsilon_1 < \epsilon_0$, it follows that w.p. 1 for $N$ large enough*

$$\sup_{\theta \in V_{\epsilon_0}} \mu_N(\theta) \leq \kappa(\epsilon_2)^{-1}e^{-N(\epsilon_1 - \epsilon_2)},$$

*where $V_{\epsilon_0}$ and $U_{\epsilon_0}$ are defined in (3.2), and $\kappa(\epsilon_2) := \int_{U_{\epsilon_2}} d\theta$.*

**Lemma B.2.** *Suppose that Assumption B.3 holds. $\forall \delta > 0, \exists \epsilon > 0$ such that $d(\theta, \Theta^*) < \delta$ for all $\theta \in U_\epsilon$.*

*Proof.* We prove this lemma by contradiction. Suppose that $\exists \delta_0 > 0$ such that $\forall \epsilon > 0$, there exists $\theta \in \Theta$ satisfying $\psi(\theta^*) - \psi(\theta) < \epsilon$ and $d(\theta, \Theta^*) \geq \delta_0$.

Choose $\epsilon = \frac{1}{n}$ and then get a sequence $\{\theta_n\}_{n=1}^\infty$. As $\Theta$ is compact, there exists a subsequence of $\{\theta_n\}_{n=1}^\infty$ that converge to a point $\theta' \in \Theta$ satisfying $d(\theta', \Theta^*) \geq \delta_0$. As $\psi$ is continuous, $\psi(\theta') = \psi(\theta^*)$. Contradiction! $\square$

Then we can prove Theorem 5

*Proof.* For any $\delta > 0$, we can choose $\epsilon_0$ such that $d(\theta, \Theta^*) \leq \delta$ for $\theta \in U_{\epsilon_0}$. Then we have

$|\rho_{\theta \sim \mu_N}(C(\alpha, \theta)) - C(\alpha, \theta^*)|$

$= |\max_{\xi:\xi \in \mathcal{U}(\mu_N)} \int_{\theta \in \Theta} \xi(\theta)\mu_N(\theta)[C(\alpha, \theta) - C(\alpha, \theta^*)]d\theta|$

$\leq \max_{\xi:\xi \in \mathcal{U}(\mu_N)} \int_{U_{\epsilon_0}} \xi(\theta)\mu_N(\theta)|C(\alpha, \theta) - C(\alpha, \theta^*)|d\theta + \max_{\xi:\xi \in \mathcal{U}(\mu_N)} \int_{V_{\epsilon_0}} \xi(\theta)\mu_N(\theta)|C(\alpha, \theta) - C(\alpha, \theta^*)|d\theta$

$\leq \sup_{\|\theta - \theta^*\| \leq \delta} |C(\alpha, \theta) - C(\alpha, \theta^*)| + 2\sup_{\theta \in \Theta}|C(\alpha^*, \theta)| \max_{\xi:\xi\mu_N \in \mathcal{U}(\mu_N)} \int_{V_\epsilon} \xi(\theta)\mu_N(\theta)d\theta$

By Holder's Inequality, we have

$$\int_{V_\epsilon} \xi(\theta)\mu_N(\theta)d\theta = \int_{V_\epsilon} \xi(\theta)\mu_N(\theta)^{1/q}\mu_N(\theta)^{1/p}d\theta$$

$$\leq \left[\int_{V_\epsilon} \xi(\theta)^q \mu_N(\theta)d\theta\right]^{1/q} \left[\int_{V_\epsilon} \mu_N(\theta)d\theta\right]^{1/p}$$

$$\leq B_q \kappa(\epsilon_2)^{-1/p} e^{-N(\epsilon_1 - \epsilon_2)/p} Vol(\Theta)^{1/p}$$

Thus

$$|\rho_{\theta \sim \mu_N}(C(\alpha, \theta)) - C(\alpha, \theta^*)| \leq \delta L_\theta + 2B_q \kappa(\epsilon_2)^{-1/p} e^{-N(\epsilon_1 - \epsilon_2)/p} Vol(\Theta)^{1/p} \sup_{\theta \in \Theta}|C(\alpha^*, \theta)|,$$

which implies $D_N \to 0$ as $N \to \infty$ since $\delta$ is arbitary.

Then we have

$$C(\alpha_N^*, \theta^*) - C(\alpha^*, \theta^*) \leq 2\delta L_\theta + 4B_q \kappa(\epsilon_2)^{-1/p} e^{-N(\epsilon_1 - \epsilon_2)/p} Vol(\Theta)^{1/p} \sup_{\theta \in \Theta}|C(\alpha^*, \theta)|,$$

where the last inequality holds if we assume $C(\alpha, \theta)$ is $L_\theta-$ Lipschitz continuous w.r.t. $\theta$. Let $N \to \infty$ and recall that $\delta$ is arbitary, we get the result. $\square$

### B.6  PROOF OF THEOREM 6

*Proof.* We assume that each $G_i(\alpha) := \rho_{\theta \sim \mu_i}(C(\alpha, \theta))$ has $L_{G,i}$ Lipschitz continuous gradient and define the gap term

$$y_{i,j} := \mathbb{E}[G_i(\alpha_{i,j}) - G_i(\alpha_i^*)]$$

By Theorem 4, we have

$$y_{i+1,t_{i+1}} \leq (1 - \epsilon)^{t_{i+1}} y_{i+1,0} + 2L_{G,i+1}\ell_\alpha D_\lambda^2 \epsilon + \frac{\sigma_\xi}{r_{i+1}L_{G,i+1}\epsilon}.$$

Then we connect $i + 1$-th episode with the previous one. Notice that it holds for any $\alpha$ that

$$G_{i+1}(\alpha) - G_{i+1}^*$$
$$= G_{i+1}(\alpha) - G_i(\alpha) + G_i(\alpha) - G_i^*$$
$$+ G_i^* - G_i(\alpha_{i+1}^*) + G_i(\alpha_{i+1}^*) - G_{i+1}(\alpha_{i+1}^*)$$
$$\leq 2D_i + 2D_{i+1} + G_i(\alpha) - G_i^*,$$

which implies

$$y_{i+1,0} \leq y_{i,t_i} + 2(D_i + D_{i+1}).$$

Thus we have

$$y_{i+1,t_{i+1}} \leq (1 - \epsilon)^{t_{i+1}} y_{i,t_i} + (1 - \epsilon)^{t_{i+1}} 2(D_i + D_{i+1}) + 2L_{G,i+1}\ell_\alpha D_\lambda^2 \epsilon + \frac{\sigma_\xi}{r_{i+1}L_{G,i+1}\epsilon}.$$

Note that $(1 - \epsilon)^{\epsilon^{-1}} \leq 1/2, \forall \epsilon \leq 1$. By choosing $t_{i+1} \geq \mathcal{O}(\epsilon^{-1}\log(\frac{D_i + D_{i+1}}{\epsilon}))$ and $r_{i+1} = \Theta(\epsilon^{-2}/L_{G,i+1})$, we can keep an error bound $\mathcal{O}(\epsilon)$ for each episode. $\square$

## C  EXAMPLES OF LOSS FUNCTION

**Example 2** (Imitation Learning). *During imitation learning, the agent learn through some demonstrations to behave similarly to an expert. One formulation is minimize the $f-$divergence between the occupancy measure of the current policy and the target occupancy measure:*

$$\min_{\pi} D_f(\lambda^{\pi}, q) = \sum_{s,a} q(s,a) f\left(\frac{\lambda^{\pi}(s,a)}{q(s,a)}\right)$$

## D  EXAMPLES OF RISK ENVELOP

**Example 3.** *[Conditional Value at Risk] First, Value-at-risk $\mathrm{VaR}_{\beta}(X)$ is defined as the $\beta$-quantile of $X$, i.e., $\mathrm{VaR}_{\beta}(X) := \inf\{t : \mathbb{P}(X \leq t) \geq \beta\}$, where the confidence level $\beta \in (0,1)$. Assuming there is no probability atom at $\mathrm{VaR}_{\beta}(X)$, CVaR at confidence level $\beta$ is defined as the mean of the $\beta$-tail distribution of $X$, i.e., $\mathrm{CVaR}_{\beta}(X) = \mathbb{E}[X \mid X \geq \mathrm{VaR}_{\beta}(X)]$. The envelope set is*

$$\mathcal{U}(\mu_N) = \{\xi \in \mathcal{Z}^* : \int_{\Theta} \xi(\theta)\mu_N(\theta)d\theta = 1, \xi(\theta) \in \left[0, \frac{1}{1-\beta}\right] a.s.\theta \in \Theta\}$$

**Example 4.** *(Mean-Upper-Semideviation of Order $p$). For $\mathcal{Z} := \mathcal{L}_p(\Theta, \mathcal{F}, \mu_N)$ and $\mathcal{Z}^* := \mathcal{L}_q(\Theta, \mathcal{F}, \mu_N)$, with $p \in [1, +\infty)$, $c \in [0,1]$ and $\mathcal{F}$ to be a $\sigma$-field on $\Theta$, consider*

$$\rho(Z) := \mathbb{E}[Z] + c\left(\mathbb{E}\left[[Z - \mathbb{E}[Z]]_+^p\right]\right)^{1/p},$$

*where $[a]_+^p = \max\{0,a\}^p$. Then the envelope set is*

$$\mathcal{U}(\mu_N) = \{\xi' \in \mathcal{Z}^* : \xi' = 1 + \xi - \mathbb{E}[\zeta], \|\xi\|_q \leq c, \xi \succeq 0\}\}.$$

More examples can be found in Section 6.3.2(Shapiro et al., 2021).

## E  POLICY GRADIENT FOR MDP WITH CVAR RISK MEASURE : A SPECIAL CASE STUDY

Here we offer an example of gradient estimator with a common coherent risk measure Conditional Value at Risk(CVaR), the definition of which can be found in Example 3. For the considered CVaR risk functional, (Hong & Liu, 2009) shows that the gradient of the CVaR risk functional can be expressed as

$$\nabla\, \mathrm{CVaR}_{\beta}(X(\alpha)) = \mathbb{E}[\nabla X(\alpha) | X(\alpha) \geq v_{\beta}(\alpha)]$$

where $v_{\beta} = v_{\beta}(\alpha) := \mathrm{VaR}_{\beta}(X(\alpha))$ for a random parameterized variable $X(\alpha)$ satisfying Assumption E.1. Unless otherwise specified, the derivative is assumed to be taken w.r.t. $\alpha$.

**Assumption E.1.** *(Assumption 1, 2, 3 (Hong & Liu, 2009)) (i) There exists a random variable $L$ with $\mathbb{E}(K) < \infty$ such that $|X(\alpha_2) - X(\alpha_1)| \leq K\|\alpha_2 - \alpha_1\|_2$ for all $\alpha_1, \alpha_2 \in W$, and $\nabla_{\alpha}X(\alpha)$ exists almost surely for all $\alpha \in W$.*

*(ii) VaR function $v_{\beta}(\alpha)$ is differentiable for any $\alpha \in W$.*

*(iii) For any $\alpha \in W, \mathbb{P}(X(\alpha) = v_{\beta}(\alpha)) = 0$.*

Assumption E.1 (i) is commonly used in path-wise derivative estimation; (ii) shows that VaR function is locally Lipschitz; (iii) requires that there is no probability atom at $VaR(X)$ and implies that $\mathbb{P}(X(\alpha) \geq v_{\beta}(\alpha)) = 1 - \beta$.

**Theorem 7.** *Suppose that Assumption E.1 holds. Then, for any $\alpha \in W$ and $\beta \in (0,1)$, the policy gradient to the objective function in equation 3 is given by:*

$$g(\alpha) = \mathbb{E}_{\theta \sim \mu_N}\left[\nabla C(\alpha, \theta) \mid C(\alpha, \theta) \geq v_{\beta}(\alpha)\right]$$

$$= \frac{1}{1-\beta}\mathbb{E}_{\theta \sim \mu_N}\left[\nabla C(\alpha, \theta)\mathbb{1}_{\{C(\alpha,\theta) \geq v_{\beta}\}}\right] \quad (23)$$

*where $\mathbb{1}_{\{\cdot\}}$ is the indicator function.*

If we apply Theorem 2 to CVaR, we will get the same result as Theorem7. To compute the gradient $g(\alpha)$, we require the cumulative value $C(\alpha, \theta)$ of policy $\pi_\alpha$ and its gradient $\nabla C(\alpha, \theta)$, value-at-risk $v_\beta$, as well as the evaluation of the expectation taken w.r.t. the posterior distribution $\mu_N$. Here we show how to use zeroth-order method instead of variational approach to estimate $\nabla_\alpha C(\alpha, \theta)$. Since there is no closed-form expression for the expectation, we estimate the gradient $g(\alpha)$ with samples $\{\theta^i\}_{i=1}^n$ generated from $\mu_N$. We construct the gradient estimator as follows:

$$\widehat{g}(\alpha) = \frac{1}{n(1-\beta)} \sum_{i=1}^n \widehat{\nabla C}(\alpha, \theta^i) \mathbb{1}_{\{\widehat{C}(\alpha, \theta^i) \geq \widehat{v}_\beta\}}. \tag{24}$$

For a fixed $\alpha$ and $\theta^i$, we first estimate the occupancy measure $\lambda^i$ by making a truncation of horizon $K$ in equation 1 with error

$$\|\widehat{\lambda}^i - \lambda^i\|_\infty \leq \epsilon_\lambda := \gamma^K/(1-\gamma) \tag{25}$$

for some $K > 0$. The cumulative value with the truncated occupancy measure $\widehat{\lambda}^i$ is denoted by $\widehat{C}(\alpha, \theta^i) = F(\widehat{\lambda}, P_{\theta^i})$. The value-at-risk estimate is $\widehat{v}_\beta := \widehat{C}(\alpha, \theta)_{\lceil n\beta \rceil:n}$, where $\widehat{C}(\alpha, \theta)_{\lceil n\beta \rceil:n}$ is the $\lceil n\beta \rceil$-th smallest quantity in $\{\widehat{C}(\alpha, \theta^i)\}_{i=1}^n$.

Here we adopt the Gaussian smoothing approach of estimating gradients from function evaluations (Nesterov & Spokoiny, 2017; Balasubramanian & Ghadimi, 2022). When there is no oracle to the first-order information or it is not efficient to calculate the gradient directly, Gaussian smoothing approach is a useful technique in zeroth-order method. Compared with finite difference method, Gaussian smoothing approach requires weaker smoothness condition of objective function. For a fixed $\alpha$ and $\theta^i$, generate $\{u^{i,j}\}_{j=1}^{m_i}$, where $u^{i,j} \sim \mathcal{N}(0, I_d)$. Then $\widehat{\nabla C}$ can be constructed as:

$$\widehat{\nabla C}(\alpha, \theta^i) = \frac{1}{m_i} \sum_{j=1}^{m_i} \frac{\widehat{C}(\alpha + \nu u^{i,j}, \theta^i) - \widehat{C}(\alpha, \theta^i)}{\nu} u^{i,j} \tag{26}$$

where $\nu > 0$ is the smoothing parameter.

For ease of notation, let $\widehat{G}(\alpha)$ denote the sample estimate of $\rho_{\theta \sim \mu_N}(C(\alpha, \theta))$. We use the following gradient descent step in the $t$-th iteration:

$$\alpha_{t+1} = \arg\min_{\alpha \in W} \widehat{G}(\alpha_t) + \langle \widehat{g}(\alpha_t), \alpha - \alpha_t \rangle + \frac{\eta_t}{2} \|\alpha - \alpha_t\|^2$$
$$= \text{Proj}_W \left( \alpha_t - \frac{1}{\eta_t} \widehat{g}(\alpha_t) \right) \tag{27}$$

where $\eta_t$ is the stepsize and $\text{Proj}_W(x) = \arg\min_{y \in W} \|y - x\|_2^2$ projects $x$ into the parameter space $W$. We summarize the full algorithm in Algorithm 4.

### E.1 CONVERGENCE ANALYSIS FOR CVAR RISK MEASURE

Here we only show the estimation error of the policy gradient. To get a finite-step convergence result similar to Theorem 4, we only need to substitute $\mathcal{O}(r_N^{-1/4})$ in Theorem 4 with $\mathcal{O}(R^{1/2})$, where $R^2 = \mathcal{O}\left(dn^{-1} + \epsilon_\lambda + \frac{d\epsilon_\lambda^2}{\nu^2} + \frac{d + \nu^2 d^3}{m}\right)$ is the bound for $\mathbb{E}\|[g - \widehat{g}]\|_2^2$ in Theorem 8.

Here we still adopt the Assumption 3.2 about the smoothness for the considered loss functions, which are commonly used in gradient descent analysis. The error bound for the zeroth-order estimation for $\nabla C$ is then shown in the next lemma.

**Lemma E.1.** *Suppose Assumption E.1 and Assumption 3.2 hold. Then we have for each $i \in [n]$*

$$\mathbb{E}\|\widehat{\nabla C}(\alpha, \theta_i) - \nabla C(\alpha, \theta_i)\|_2^2 \leq \frac{8d}{\nu^2} L_{F,\infty}^2 \epsilon_\lambda^2$$
$$+ \frac{8(d+5)B^2}{m_i} + \frac{2\nu^2 L_{C,2}^2 (d+6)^3}{m_i}, \tag{28}$$

*where $L_{F,\infty}, L_{C,2}, B$ are constants in Assumption 3.2, $\epsilon_\lambda$ is the truncation error defined in equation 25, $d$ is the dimension of the policy parameter $\alpha$, $m_i$ is the number of samples used to construct the zeroth-order estimator in equation 26.*

---

**Algorithm 4** BR-PG: Bayesian Risk Policy Gradient for CVaR

---

**input**: initial $\alpha_0$, data $\zeta^{(N)}$ of size $N$, prior distribution $\mu_0(\theta)$, iteration number $T$, truncation horizon $K$;

calculate the posterior $\mu_N(\theta) = \frac{P_\theta(\zeta^{(N)})\mu_0(\theta)}{\int_{\theta'} P_{\theta'}(\zeta^{(N)})\mu_0(\theta')}$;

**for** $t = 0$ to $T - 1$ **do**

    sample $\{\theta_t^i\}_{i=1}^n$ from $\mu_N(\theta)$;

    **for** $i = 1$ to $n$ **do**

        calculate $\widehat{\lambda}_t^i$ using the truncation of horizon $K$ specified in equation 1;

        calculate $\widehat{C}(\alpha_t, \theta_t^i) := F(\widehat{\lambda}_t^i, P_{\theta_t^i})$;

        generate $\{u^{i,j}\}_{j=1}^{m_i}$, where $u^{i,j} \sim \mathcal{N}(0, I_d)$;

        calculate $\widehat{\nabla C}(\alpha_t, \theta_t^i)$ by equation 26;

    **end for**

    calculate $\widehat{v}_\beta(\alpha_t) := \widehat{C}(\alpha_t, \theta_t^i)_{\lceil n\beta\rceil:n}$.

    calculate $\widehat{g}(\alpha_t)$ by equation 24;

    update $\alpha_{t+1}$ by equation 10.

**end for**

**output**: $\alpha_T$.

---

**Assumption E.2.** *(Assumptions 4 and 5 in (Hong & Liu, 2009)*

*(1) For all $\alpha \in W$, $C(\alpha, \theta)$ is a continuous random variable with a density function $f_{C,\alpha}(y)$. Furthermore, $f_{C,\alpha}(y)$ and $g_{C,\alpha}(y) := \mathbb{E}_\theta[\nabla C(\alpha, \theta) \mid C(\alpha, \theta) = y]$ are continuous at $y = v_\alpha$, and $f_{C,\alpha}(v_\alpha) > 0$.*

*(2) $\mathbb{E}_\theta\left[C(\alpha, \theta)^2\right] < \infty$ for all $\alpha \in W$.*

Now we are ready to show the error for our gradient estimator given in equation 24.

**Theorem 8.** *Suppose that Assumption E.1, Assumption 3.2 and Assumption E.2 hold. Also assume that the cumulative distribution function of $C(\alpha, \theta)$ w.r.t $\theta$ is $\ell_C-$ Lipschitz continuous for each $\alpha \in W$. Let $m_i = m \ \forall i \in [n]$. Then for each $\alpha \in W$,*

$$\mathbb{E}\|[g - \widehat{g}]\|_2^2 \leq \mathcal{O}\left(dn^{-1} + \epsilon_\lambda + \frac{d\epsilon_\lambda^2}{\nu^2} + \frac{d + \nu^2 d^3}{m}\right),$$

*where $n$ is the number of samples of $\theta$.*

*Proof.* First recall that the true gradient and our gradient estimator are $g = \frac{1}{1-\beta}\mathbb{E}\left[\nabla C(\alpha, \theta)\mathbb{1}_{\{C(\alpha,\theta)\geq v_\beta\}}\right]$ and $\widehat{g} = \frac{1}{n(1-\beta)}\sum_{i=1}^n \widehat{\nabla C}(\alpha, \theta_i)\mathbb{1}_{\{\widehat{C}(\alpha,\theta_i)\geq\widehat{v}_\beta\}}$. Let

$$\tilde{g} = \frac{1}{n(1-\beta)}\sum_{i=1}^n \nabla C(\alpha, \theta_i)\mathbb{1}_{\{C(\alpha,\theta_i)\geq\tilde{v}_\beta\}},$$

and

$$\widehat{g_1} = \frac{1}{n(1-\beta)}\sum_{i=1}^n \nabla C(\alpha, \theta_i)\mathbb{1}_{\{\widehat{C}(\alpha,\theta_i)\geq\widehat{v}_\beta\}},$$

where $\tilde{v}_\beta := C(\alpha, \theta_i)_{\lceil n\beta\rceil:n}$. Then we have the decomposition $g - \widehat{g} = (g - \tilde{g}) + (\tilde{g} - \widehat{g_1}) + (\widehat{g_1} - \widehat{g}) := R_1 + R_2 + R_3$. For $R_1$, it is the error in the estimation of expectation taken w.r.t. $\theta$. Suppose that Assumption E.1 and Assumption E.2 hold, Theorem 4.2 from (Hong & Liu, 2009) shows that

$$\|\mathbb{E}R_1\|_2 = \|\mathbb{E}[\tilde{g}] - g\|_2 = o(n^{-1/2}d^{-1/2}).$$

Notice that

$$\|g - \tilde{g}\|_2^2 \leq 2\|g - \mathbb{E}\tilde{g}\|_2^2 + 2\|\mathbb{E}\tilde{g} - \tilde{g}\|_2^2.$$

By Theorem 4.3 from (Hong & Liu, 2009), $Var(\tilde{g}) = \mathcal{O}(dn^{-1})$. Thus

$$\mathbb{E}\|R_1\|_2^2 = \mathcal{O}(dn^{-1}). \tag{29}$$

For $R_3$, it is the error in the estimation of $C(\alpha, \theta)$. By Lemma E.1, $\mathbb{E}[\|\widehat{\nabla C}(\alpha, \theta_i) - \nabla C(\alpha, \theta_i)\|_2^2] \leq \frac{8d}{\nu^2} L_{F,\infty}^2 \epsilon_\lambda^2 + \frac{8(d+5)B^2}{m_i} + \frac{2\nu^2 L_{C,2}^2 (d+6)^3}{m_i}$. If we choose all $m_i$ to be the same $m$, then

$$\mathbb{E}[\|\widehat{g_1} - \widehat{g}\|_2^2] \leq \frac{1}{n(1-\beta)^2} \sum_{i=1}^n \|\widehat{\nabla C}(\alpha, \theta_i) - \nabla C(\alpha, \theta_i)\|_2^2$$

$$\leq \mathcal{O}\left(\frac{d\epsilon_\lambda^2}{\nu^2} + \frac{d+5}{m} + \frac{\nu^2(d+6)^3}{m}\right).$$

Thus

$$\mathbb{E}[\|R_3\|_2^2] \leq \mathcal{O}\left(\frac{d\epsilon_\lambda^2}{\nu^2} + \frac{d+5}{m} + \frac{\nu^2(d+6)^3}{m}\right). \tag{30}$$

Now we consider $R_2$. Define the event $A_i = \{C(\alpha, \theta_i) \geq \widetilde{v}_\beta\}, \widehat{A_i} = \{\widehat{C}(\alpha, \theta_i) \geq \widehat{v}_\beta\}$ and $A_i \Delta \widehat{A_i} := (A_i \backslash \widehat{A_i}) \cup (\widehat{A_i} \backslash A_i)$. Then

$$\|R_2\|_2 \leq \frac{1}{n(1-\beta)} \sum_{i=1}^n \|\nabla C(\alpha, \theta_i)\|_2 \cdot \mathbb{1}_{A_i \Delta \widehat{A_i}}$$

$$\leq \frac{1}{n(1-\beta)} \sum_{i=1}^n B \mathbb{1}_{A_i \Delta \widehat{A_i}},$$

and

$$\|R_2\|_2^2 \leq \frac{1}{n^2(1-\beta)^2} \left(\sum_{i=1}^n B \mathbb{1}_{A_i \Delta \widehat{A_i}}\right)^2$$

$$\leq \frac{1}{n(1-\beta)^2} B^2 \sum_{i=1}^n \mathbb{1}_{A_i \Delta \widehat{A_i}}.$$

Notice that

$$\mathbb{P}(\mathbb{1}_{A_i \Delta \widehat{A_i}}) = \mathbb{P}(A_i \backslash \widehat{A_i}) + \mathbb{P}(\widehat{A_i} \backslash A_i).$$

As the estimation error of $\lambda$, i.e. $\|\hat{\lambda} - \lambda\|_\infty$, is bounded by $\epsilon_\lambda$ and $F$ is $L_{F,\infty}$-Lipschitz continuous w.r.t $\|\cdot\|_\infty$, we have $|\widehat{C}(\alpha, \theta_i) - C(\alpha, \theta_i)| \leq L_{F,\infty} \epsilon_\lambda$. As a result, $|\widetilde{v}_\beta - \widehat{v}_\beta| \leq L_{F,\infty} \epsilon_\lambda$. Notice that $\{C(\alpha, \theta_i) \geq \widetilde{v}_\beta + 2L_{F,\infty} \epsilon_\lambda\} \subseteq \{\widehat{C}(\alpha, \theta_i) \geq \widehat{v}_\beta\} \subseteq \{C(\alpha, \theta_i) \geq \widetilde{v}_\beta - 2L_{F,\infty} \epsilon_\lambda\}$. Then we have $\mathbb{P}(A_i \backslash \widehat{A_i}) + \mathbb{P}(\widehat{A_i} \backslash A_i) \leq 4\ell_C L_{F,\infty} \epsilon_\lambda$, by the assumption on the cumulative distribution function of $C$, and thus

$$\mathbb{E}\|R_2\|_2^2 \leq \frac{4}{(1-\beta)^2} B^2 \ell_C L_{F,\infty} \epsilon_\lambda = \mathcal{O}(\epsilon_\lambda). \tag{31}$$

Combining equation 29, equation 30 and equation 31, we have

$$\mathbb{E}\|[g - \widehat{g}]\|_2^2 \leq \mathcal{O}\left(dn^{-1} + \epsilon_\lambda + \frac{d\epsilon_\lambda^2}{\nu^2} + \frac{d + \nu^2 d^3}{m}\right).$$

$\square$

Theorem 8 implies that the error of the gradient estimator can be reduced to arbitrarily small by increasing the sample size $n, m$ or decreasing the truncation error $\epsilon_\gamma$.

# F    IMPLEMENTING DETAILS

**Frozen lake problem**. Consider moving from the Start (S) to the Goal (G) on an $5 \times 5$ frozen lake with 6 holes (H). Then there are 18 ices (F) (involving Start). The agent may not move in the intended direction as the ice is slippery. The position is the row-column coordinate $(i, j)$ with $i, j \in \{0, 1, 2, 3, 4\}$ and the state is the $5 * i + j$. The state space is $\{0, 1, \ldots, 24\}$. The action set consists of moving in four directions. The unknown slippery probability is $\theta_s$. Before reaching the goal and standing on the ice, the agent may move in the intended direction with unknown probability

$1 - \theta_s$ and move in either perpendicular direction with probability $\theta_s/2$. When falling into the hole, the agent may try to escape from the hole and move to the intended direction. Each time the agent will succeed in escaping from the hole with unknown probability $\theta_e$. After reaching the Goal, the agent will always stay in the Goal whatever the action is. We set the cost to be 1 for each action on ice before reaching goal. Also, stronger efforts may be made when it is harder to escape from the hole. So we set the per-action cost in hole to be uniformly distributed between $[1, 1+2(1-\theta_e)]$. We aim to find a policy with the minimum general loss function. The data set consists of $N$ historical slippery movements and escapement trials.

**Linear Loss.** For each of the considered formulations, we obtain the corresponding optimal policy for the same data set and evaluate the actual performance of the obtained policy on the true system, i.e. MDP with the true parameter $\theta^c$. Specifically, we use the linear loss function, which corresponds to the total discounted cost in a classical MDP problem. This is referred to as one replication, and we repeat the experiments for 50 replications using different independent data sets. Results for the frozen lake problem are presented in Table 1, with varying data size $N = 5$ and $N = 50$, slippery probability $\theta_s = 0.3$ and escape probability $\theta_e = 0.02$. Note that we report the positive-sided variance, which corresponds to the second order moment of the positive component of the difference between the actual loss and the expected loss. Intuitively, a high positive-sided variance indicates more replications with higher costs than the average, which is undesirable.

**Episodic Case**. We consider the episodic setting where the data collection and policy update are alternatively conducted. Similar with the previous case with fixed data size, we consider the mean loss function with slippery probability $\theta_s = 0.3$, escape probability $\theta_e = 0.02$, and $5 \times 20$, $10 \times 10$, $20 \times 5$ iterations in total. We repeat the experiments for 50 replications on different independent data sets. Figure 1 shows the decrease of the loss function by different methods.

Results for the frozen lake problem with escape probability $\theta_e = 0.7$ can be found in Table 2 and Table 3.

Table 2: Results for frozen lake problem. Expected loss and positive-sided variance at different risk levels $\alpha$ are reported for different algorithms. Standard errors are reported in parentheses. Escape probability $\theta_e = 0.7$ and number of data points is $N = 5$.

| Approach | loss function: mean | |
|---|---|---|
| | expected loss | positive-sided variance |
| BR-PG ($\beta = 0$) | 10.322 (0.0182) | 0.0153 |
| BR-PG ($\beta = 0.5$) | 10.520(0.105) | 0.502 |
| BR-PG ($\beta = 0.9$) | 11.718 (0.357) | 4.982 |
| Empirical | 11.667 (0.0687) | 0.156 |
| DRQL (radius=0.05) | 11.223(0.185) | 1.283 |
| DRQL (radius=1) | 20.751(1.438) | 69.514 |
| DRQL (radius=20) | 23.181(1.396) | 57.495 |

Table 3: Results for frozen lake problem. Expected loss and positive-sided variance at different risk levels $\alpha$ are reported for different algorithms. Standard errors are reported in parentheses. Escape probability $\theta_e = 0.7$ and number of data points is $N = 50$.

| Approach | loss function: mean | |
|---|---|---|
| | expected loss | positive-sided variance |
| BR-PG ($\beta = 0$) | 10.271 (0.00227) | 0.000197 |
| BR-PG ($\beta = 0.5$) | 10.256 (0.00211) | 0.000188 |
| BR-PG ($\beta = 0.9$) | 10.230(0.00294) | 0.000398 |
| Empirical | 11.316 (0.0235) | 0.017 |
| DRQL (radius=0.05) | 10.888( 0.171) | 1.235 |
| DRQL (radius=1) | 20.990( 1.324) | 56.027 |
| DRQL (radius=20) | 23.500(1.282) | 51.915 |

Figure 3 shows the map of the frozen lake problem with 1 Start(S), 1 Goal(G), 6 holes(H) and remaining frozen(F) parts. We design such a map so that the agent has to avoid falling in the hole when the escape probability is very small and cross the hole when the escape probability is high. Detailed parameters are set as follows. The true slippery probability is 0.3. The iteration number for gradient descent is 100, the stepsize is 0.5, and the sample number in each iteration is $r_N = 30$. we set the discounter factor to be $\gamma = 0.97$, the truncation horizon for occupancy measure to be $K = 130$. equation 26.

For the "mean" loss function, we use the maximum likelihood estimator (MLE) of $\theta$ as the empirical measure to be compared with BR-PG. Also, we use the distributionally robust Q-learning (DRQL)(Liu et al., 2022) with different radius for the KL divergence ball as another benchmark. We also use the MLE of $\theta$ as the parameter for the center of the KL divergence ball in DRQL with different radius. For BR-PG, the sample number from posterior in each iteration is 30, the total iteration number is 100, the step size of SGD is chosen to be 1, and the prior distributions are chosen to be $\text{Beta}(1,1)$ for two parameters. We show the histogram of total cost over 50 replications for all methods in Figure 4 with the risk level 0.8 for CVaR over replications, which visualize the measures of dispersion.

**Mimicking a policy**. Here we consider a different problem of mimicking an expert policy still using Frozen Lake environment. Given an expert policy, we have access to the state distribution of the expert policy under the true environment, which is denoted by a nonnegative function $J$ satisfying $\sum_{s \in \mathcal{S}} J(s) = 1$. The loss function we want to minimize is defined as the KL divergence between state occupancy measure under the current policy and the expert state distribution $F(\lambda) = \text{KL}\left((1-\gamma)\sum_{a \in \mathcal{A}} \lambda_a \| J\right) = \sum_{s \in \mathcal{S}}\sum_{a \in \mathcal{A}}(1-\gamma)\lambda_{sa} \log\left(\frac{\sum_{a \in \mathcal{A}}(1-\gamma)\lambda_{sa}}{J(s)}\right)$. We compare the BR-PG algorithm with CVaR risk measure under different risk levels $\beta = 0, 0.5, 0.9$, respectively, with the benchmark empirical approach using the MLE estimator for the parameter as before. Figure 2 shows the decrease of the loss function by different methods. It should be noticed that DRQL can only be applied to the "mean" loss function, thus we don't use it as a benchmark. The performance of the 50 replications is shown in figure 5, where the shown results start from the 30-th iteration.

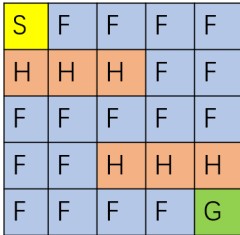

Figure 3: Map of frozen lake problem

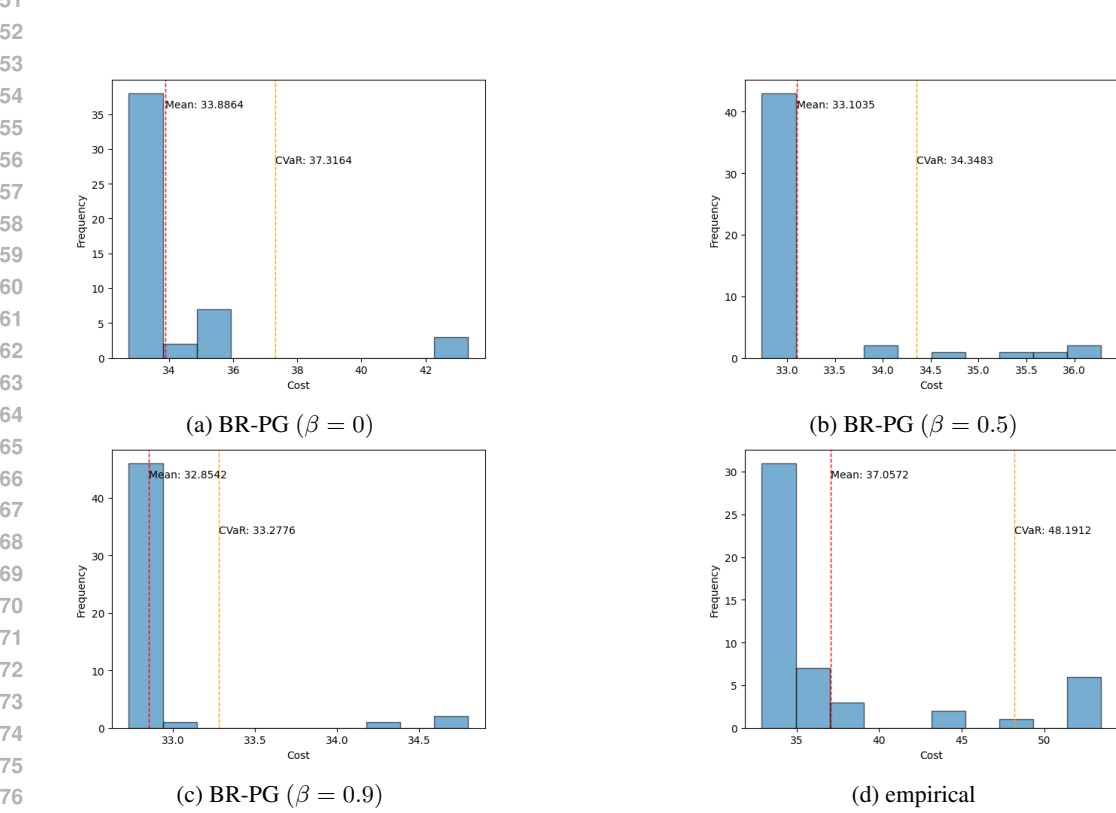

(a) BR-PG ($\beta = 0$)

(b) BR-PG ($\beta = 0.5$)

(c) BR-PG ($\beta = 0.9$)

(d) empirical

Figure 4: Result for utility function "mean" with data size $N = 5$ and escape probability $\theta_e = 0.02$

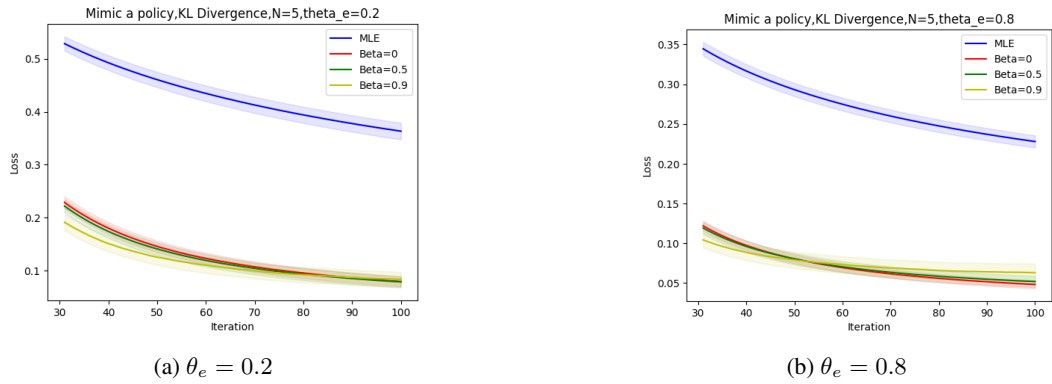

(a) $\theta_e = 0.2$

(b) $\theta_e = 0.8$

Figure 5: Results for utility function "KL divergence" with data size $N = 5$ and escape probability $\theta_e = 0.2$ and $\theta_e = 0.8$

