# OpenReview forum: "Policy Gradient Optimization for Markov Decision Processes with Epistemic Uncertainty and General Loss Functions"
_ICLR.cc/2025/Conference — ICLR 2025 Conference Withdrawn Submission_

### Official Review · Reviewer_adYX · 2024-10-26

**Soundness:** 4
**Presentation:** 4
**Contribution:** 4
**Rating:** 8
**Confidence:** 3

**Summary:**

In this paper, the authors build upon the work presented in DRQL (Zhang et al.) and the Envelope Theorem. They propose a novel algorithm aimed at managing environmental uncertainties, specifically targeting variations in the transition kernel. The proposed approach leverages offline training for model stability, which is subsequently tested online, thus addressing real-world dynamic uncertainties effectively. Additionally, the authors extend the method's application from discrete to continuous cases, demonstrated through OpenAI benchmark scenarios.

**Strengths:**

1. Novel Approach to Handling Environmental Uncertainty: The proposed method presents an innovative solution to managing uncertainty in dynamic environments, particularly useful in areas requiring robustness against fluctuating data distributions.
2. Solid Theoretical Foundation and Proof: The paper showcases a rigorous theoretical basis, with clear proofs supporting the stability and convergence of the method, adding confidence in the reliability of the algorithm.
3. Impressive Comparison Results in the Case Study: The authors provide compelling results from case study comparisons, particularly illustrating the method's advantages over traditional approaches in practical applications.

**Weaknesses:**

1. Additional Comparative Case Study: Including a comparative case study in a different environment could strengthen the evaluation by illustrating the method's adaptability across various domains. For example, assessing performance in a stochastic, multi-agent environment could provide insight into how the method handles complex dynamics beyond the controlled setup, such as the autonomous driving scenario.

2. Incorporation of a Real-World Dataset: Applying the algorithm to a specific real-world dataset, if feasible, could highlight its practical relevance and capability to handle real-life uncertainties. For instance, testing on a dataset related to robotics, such as sensor-based navigation or autonomous vehicle trajectory data, might illustrate how effectively the algorithm adapts to real-world conditions.

3. Comparison with Assumptive Environments and BR-PG Method: A comparative analysis with assumed environments, along with the BR-PG method the authors propose, would provide a fuller understanding of the algorithm’s performance relative to established techniques.

**Questions:**

Question: Could the authors elaborate on how the transition kernel adapts in real-world settings, particularly in dynamically changing conditions? Specifically, it would be insightful to understand how the episodic setting in Algorithm 2 might be adapted to handle evolving environments, such as those encountered in autonomous vehicle navigation. This could provide practical insights into the algorithm's adaptability and robustness in non-static scenarios.

Suggestion: It would be beneficial if the authors could elaborate on the model's practical utility in real-life applications like autonomous vehicles. Specifically, understanding how the transition kernel adapts to evolving environmental factors (e.g., weather, road conditions, traffic) and whether the model supports continuous real-time updates would offer a clearer picture of its suitability for such dynamic and safety-critical domains. Real-world examples or simulations of this adaptive capability could greatly enhance the relevance and applicability of the proposed approach.

---

> ### Author Response · Authors · 2024-11-22
>
> **Comment**:
>
> Additional Comparative Case Study: Including a comparative case study in a different environment could strengthen the evaluation by illustrating the method's adaptability across various domains. For example, assessing performance in a stochastic, multi-agent environment could provide insight into how the method handles complex dynamics beyond the controlled setup, such as the autonomous driving scenario.
>
> **Response**:
>
> Thank you for your comment and suggestion. Exploring a different environment, such as a stochastic or multi-agent setting, is indeed a promising direction for future research. In multi-agent reinforcement learning (RL), each agent would have its own posterior estimation, often based on assumptions such as global state observability, which will require the fusion of the individual posteriors before optimization. So, a different approach will be needed for the multi-agent environment. While this is an excellent idea, we believe it is beyond the scope of the current work, as we have already addressed a substantial number of challenges in this conference paper. We consider this a valuable direction for a future study.
>
> **Comment**:
>
> Incorporation of a Real-World Dataset: Applying the algorithm to a specific real-world dataset, if feasible, could highlight its practical relevance and capability to handle real-life uncertainties. For instance, testing on a dataset related to robotics, such as sensor-based navigation or autonomous vehicle trajectory data, might illustrate how effectively the algorithm adapts to real-world conditions.
>
> **Response**:
>
> Thank you for the great suggestion.
>    We agree it is a good idea to demonstrate our algorithm on a real-world dataset. If time permits, we will add such an example to the paper. Our current numerical experiments focus on comparison with other benchmark methods, so as to facilitate the understanding of why and how our algorithm works well.
>
> **Comment**:
>
> Comparison with Assumptive Environments and BR-PG Method: A comparative analysis with assumed environments, along with the BR-PG method the authors propose, would provide a fuller understanding of the algorithm’s performance relative to established techniques.
>
> **Response**:
>
> Thank you for the great suggestion. We followed your suggestion to test BR-PG under the assumed environments (i.e., using the true transition probabilities in the problem) and then use  the  value iteration method to solve the true MDP. The optimal cost is 32.499, which is close to  our BR-PG method, implying BR-PG works well even if it does not know the true transition probabilities. We will update our paper using this new result. This table below is a brief summary of results.
>
>
> | Approach | N=5 |N=50|
> | -------- | ------- |-------- |
> | Approach  | linear loss | linear loss |
> | BR-PG ($\beta = 0$) |33.886 |32.784|
> | BR-PG ($\beta = 0.5$)  | 33.104|32.757|
> |  BR-PG ($\beta = 0.9$)  | 32.854 |32.741|
> | True Model  | 32.499 |32.499 |
>
> **Comment**:
>
> Question: Could the authors elaborate on how the transition kernel adapts in real-world settings, particularly in dynamically changing conditions? Specifically, it would be insightful to understand how the episodic setting in Algorithm 2 might be adapted to handle evolving environments, such as those encountered in autonomous vehicle navigation. This could provide practical insights into the algorithm's adaptability and robustness in non-static scenarios.
>
>   Suggestion: It would be beneficial if the authors could elaborate on the model's practical utility in real-life applications like autonomous vehicles. Specifically, understanding how the transition kernel adapts to evolving environmental factors (e.g., weather, road conditions, traffic) and whether the model supports continuous real-time updates would offer a clearer picture of its suitability for such dynamic and safety-critical domains. Real-world examples or simulations of this adaptive capability could greatly enhance the relevance and applicability of the proposed approach.
>
> **Response**:
>
> Thank you for your comment and question. For a dynamically changing environment, or a non-stationary MDP, the transition kernel may be time-varying. In this case, we  need the posterior estimation to be time-varying, and it is where episodic setting may work. For example, when we  collect new data in a new episode, one way is to estimate a new posterior and use some techniques, such as a moving average or exponentially forgetting weight average of the posteriors over time, to train the policy. Then the agent can use the weighted posterior to update the policy.  It is one prospective future direction to consider time-varying or multi-agent environment, but it is out of the scope of this paper.

---

> ### Author Response · Authors · 2024-11-23
>
> We sincerely appreciate your thoughtful feedback and have carefully incorporated your suggestions into the revised paper, with all changes highlighted in blue for your convenience.
>
> In response to your comments, we have added a  comparison with assumptive environments and the BR-PG method. We hope these additions address your concerns and further enhance the clarity and quality of the paper.

---

> > ### Comment · Reviewer_adYX · 2024-11-27
> >
> > Thank you for your response. I will maintain my score.

---

### Official Review · Reviewer_dfLD · 2024-10-27

**Soundness:** 2
**Presentation:** 3
**Contribution:** 2
**Rating:** 6
**Confidence:** 4

**Summary:**

The paper studies the policy gradient optimization problem for MDP with epistemic uncertainty and convex in occupancy measures. It first uses the dual representation to establish the policy gradient for this general objective. Then, the paper shows the O(1/t) convergence rate of the policy gradient algorithm, up to a gradient error term. The paper also extends the analysis to the episodic setting, providing bounds on the number of iterations required for achieving the desired accuracy. Finally, the paper uses numerical experiments to demonstrate the effectiveness of the algorithm.

**Strengths:**

* The paper develops rigorous theorems for both the infinite-discounted setting and the episodic setting.
* The paper provides numerical examples to illustrate the effectiveness of the algorithm, and the code is also released.

**Weaknesses:**

* I think the paper lacks a comprehensive review of related literature. For example, the considered problem is closely related to general-utility RL (or convex RL), yet the only cited paper is (Zhang et al., 2020). Beyond, the cited paper, there are still many relevant works, such as [1-6] below. I suggest the author includes an additional section to discuss the related literature, including previous research on convex RL.
* The introduction of the problem formulation can be more clear. Specifically, the author can emphasize/reiterate that Eq (2) specifies the problem that the paper focuses on. I took a while to figure out research problem of the paper.
* I think the author should not ignore the sample complexity in the theoretical analysis. For example, in Theorem 4, the complexity indeeds comes from the gradient estimation error term. I think a better presentation can be, discuss the iteration complexity in the exact setting (1/t), and discuss the sample complexity in the sample based setting (seems to be 1/epsilon^4).



[1] Zhang, J., Ni, C., Szepesvari, C., & Wang, M. (2021). On the convergence and sample efficiency of variance-reduced policy gradient method. Advances in Neural Information Processing Systems, 34, 2228-2240.

[2] Zhang, J., Bedi, A. S., Wang, M., & Koppel, A. (2022, June). Multi-agent reinforcement learning with general utilities via decentralized shadow reward actor-critic. In Proceedings of the AAAI Conference on Artificial Intelligence (Vol. 36, No. 8, pp. 9031-9039).

[3] Ying, D., Guo, M. A., Ding, Y., Lavaei, J., & Shen, Z. J. (2023, June). Policy-based primal-dual methods for convex constrained markov decision processes. In Proceedings of the AAAI Conference on Artificial Intelligence (Vol. 37, No. 9, pp. 10963-10971).

[4] Bai, Q., Bedi, A. S., Agarwal, M., Koppel, A., & Aggarwal, V. (2023). Achieving zero constraint violation for concave utility constrained reinforcement learning via primal-dual approach. Journal of Artificial Intelligence Research, 78, 975-1016.

[5] Barakat, A., Fatkhullin, I., & He, N. (2023, July). Reinforcement learning with general utilities: Simpler variance reduction and large state-action space. In International Conference on Machine Learning (pp. 1753-1800). PMLR.

[6] Ying, D., Zhang, Y., Ding, Y., Koppel, A., & Lavaei, J. (2024). Scalable primal-dual actor-critic method for safe multi-agent rl with general utilities. Advances in Neural Information Processing Systems, 36.

**Questions:**

* The overall flow of the paper is very similar to (Zhang et al., 2020). Could the author detailedly discuss the distinctions of the two works, and the new challenges in this more general formulation?
* In literature [2,3,5,6] mentioned above, they all use a relaxed assumption on the policy parameterization, i.e., a locally invertible mapping instead of a universally invertible mapping. Could the results in this paper also extend to the relaxed assumption?

Besides the two questions above, I hope author also responds to the weakness that I mentioned above. If these problems can be addressed, I am willing to reconsider the rating.

---

> ### Author Response · Authors · 2024-11-22
>
> **Comment**:
>
> I think the paper lacks a comprehensive review of related literature. For example, the considered problem is closely related to general-utility RL (or convex RL), yet the only cited paper is (Zhang et al., 2020). Beyond, the cited paper, there are still many relevant works, such as [1-6] below. I suggest the author includes an additional section to discuss the related literature, including previous research on convex RL.
>
> **Response**:
>
> Thank you for your comment and question.
> We will add the following paragraph to the introduction of the paper:
>
> ``Our problem is highly relevant to convex RL, which generalizes cumulative reward on a convex general-utility objective instead of cumulative reward.
> Specifically, our problem is closely tied to convex RL, which extends the traditional cumulative reward framework to a convex general-utility objective. Prior research has explored policy gradient methods to address convex RL. For instance, [7] demonstrates that the policy gradient of convex RL can be formulated as a min-max optimization problem. To reduce estimator variance, [1] introduces an off-policy policy gradient estimator that leverages mini-batch techniques and truncation mechanisms, while [5] employs a recursive approach to handle large state-action spaces. In the domain of multi-agent convex RL, [2] assumes global state observability and proposes a trajectory-based actor-critic method. Recent studies have also focused on safe convex RL, where the objective is to maximize a convex utility function under convex safety constraints. For example, [3] develops a primal-dual algorithm with strong guarantees on the optimality gap and constraint violations, achieving an
> $\mathcal{O}(1/\epsilon^3)$
>  bound in the convex-concave case with zero constraint violation. Building on this, [4] improves the bound to
> $\mathcal{O}(1/\epsilon^2)$. Furthermore, [6] extends the primal-dual framework to multi-agent convex safe RL."
>
> [1] Zhang, J., Ni, C., Szepesvari, C.,  Wang, M. (2021). On the convergence and sample efficiency of variance-reduced policy gradient method. Advances in Neural Information Processing Systems, 34, 2228-2240.
>
> [2] Zhang, J., Bedi, A. S., Wang, M.,  Koppel, A. (2022, June). Multi-agent reinforcement learning with general utilities via decentralized shadow reward actor-critic. In Proceedings of the AAAI Conference on Artificial Intelligence (Vol. 36, No. 8, pp. 9031-9039).
>
> [3] Ying, D., Guo, M. A., Ding, Y., Lavaei, J.,  Shen, Z. J. (2023, June). Policy-based primal-dual methods for convex constrained markov decision processes. In Proceedings of the AAAI Conference on Artificial Intelligence (Vol. 37, No. 9, pp. 10963-10971).
>
> [4] Bai, Q., Bedi, A. S., Agarwal, M., Koppel, A.,  Aggarwal, V. (2023). Achieving zero constraint violation for concave utility constrained reinforcement learning via primal-dual approach. Journal of Artificial Intelligence Research, 78, 975-1016.
>
> [5] Barakat, A., Fatkhullin, I.,  He, N. (2023, July). Reinforcement learning with general utilities: Simpler variance reduction and large state-action space. In International Conference on Machine Learning (pp. 1753-1800). PMLR.
>
> [6] Ying, D., Zhang, Y., Ding, Y., Koppel, A.,  Lavaei, J. (2024). Scalable primal-dual actor-critic method for safe multi-agent rl with general utilities. Advances in Neural Information Processing Systems, 36.
>
> [7] Zhang J, Koppel A, Bedi A S, et al. Variational policy gradient method for reinforcement learning with general utilities[J]. Advances in Neural Information Processing Systems, 2020, 33: 4572-4583.
>
> **Comment**:
>
> The introduction of the problem formulation can be more clear. Specifically, the author can emphasize/reiterate that Eq (2) specifies the problem that the paper focuses on. I took a while to figure out research problem of the paper.
>
> **Response**:
>
> Thank you for your comment and question.  We follow
> your suggestion and add one sentence "We aim to solve problem (2) in this paper." after Eq (2).

---

> ### Author Response · Authors · 2024-11-22
>
> **Comment**:
>
> I think the author should not ignore the sample complexity in the theoretical analysis. For example, in Theorem 4, the complexity indeeds comes from the gradient estimation error term. I think a better presentation can be, discuss the iteration complexity in the exact setting (1/t), and discuss the sample complexity in the sample based setting (seems to be $1/\epsilon^4$).
>
> **Response**:
>
> Thank you for your comment and question.  We follow
> your suggestion and revise the manuscript as follows: Theorem 4 shows the optimality gap of the objective value consisting of two parts: an asymptotically diminishing error bound $\mathcal{O}(1/t)$ in the exact setting and an estimation error bound of the policy gradient. The samples are from the posterior $\mu_N$ and the total sample complexity is $\mathcal{O}(1/\epsilon^5)$.
>
> **Comment**:
>
> The overall flow of the paper is very similar to (Zhang et al., 2020). Could the author detailedly discuss the distinctions of the two works, and the new challenges in this more general formulation?
>
> **Response**:
>
> Thank you for your comment and question. We have briefly discussed the differences between the work of Zhang et al. (2020) and our approach in the introduction. More specifically, Zhang et al. (2020) address an online convex reinforcement learning (RL) problem and derive the variational policy gradient theorem with a global convergence guarantee in the exact setting. Their gradient estimator is constructed using trajectory sampling. In contrast, we focus on an offline planning problem in a Markov Decision Process (MDP) with unknown transition probabilities, which are estimated from a fixed batch of data. Therefore, the primary difference lies in the problem setting, which introduces additional challenges, particularly with respect to risk measures. Risk measures are functions of the distribution of a random variable, and as such, the chain rule cannot be directly applied to them. Furthermore, our gradient estimator is constructed by sampling from the posterior distribution over the environment's parameters, rather than by sampling trajectories from the true environment. Additionally, the variational method used by Zhang et al. (2020) serves as a plug-in approach for our gradient estimator; however, we can also utilize other methods, such as the zeroth-order method, which is demonstrated as a special case study in Appendix E.
>
> **Comment**:
>
> In literature [2,3,5,6] mentioned above, they all use a relaxed assumption on the policy parameterization, i.e., a locally invertible mapping instead of a universally invertible mapping. Could the results in this paper also extend to the relaxed assumption?
>
> **Response**:
>
> Thank you for pointing out the relaxed assumption and the great suggestion for extending our results!  Our proof of convergence analysis use the assumption about global invertible mapping and adapts the proof from  (Zhang et al., 2020), which only consider the exact gradient setting. [2]  extends (Zhang et al., 2020) to a locally invertible mapping. After carefully checking the proof from [2], we believe our results can be extended to the relaxed assumption within a similar proof framework, and the total sample complexity can be reduced to  $\mathcal{O}(\epsilon^{-3}\log(\epsilon^{-1}))$. We will revise our paper using this relaxed assumption and updated sample complexity results.

---

> > ### Comment · Reviewer_dfLD · 2024-11-22
> >
> > Thank you very much for the responses and further clarifications. I am glad to adjust my rating if the promised revisions can be implemented.

---

> > > ### Author Response · Authors · 2024-11-23
> > >
> > > Thank you for your thoughtful feedback and re-rating. We have carefully considered your suggestions and uploaded the revised paper, with all changes highlighted in blue for your convenience.
> > >
> > > In response to your comments, we have added a comprehensive review of related literature. Additionally, the convergence rate in Theorem 4 has been improved from $1/t$ to $(1-\epsilon)^t$, and the episodic error bound in Theorem 6 has been improved from a constant to
> > > $\mathcal{O}(\epsilon)$. We hope these updates address your concerns and enhance the clarity and quality of the paper.

---

### Official Review · Reviewer_tKe3 · 2024-11-04

**Soundness:** 2
**Presentation:** 2
**Contribution:** 3
**Rating:** 6
**Confidence:** 2

**Summary:**

The authors consider a  Bayesian risk formulation for MDP. The novel loss function considers both the epistemic uncertainty about the transition kernel and the performance measure of the policy. The policy gradients are estimated using the solution of a general coherent risk measure. In the offline setting, as the number of data points reaches infinity, the propose algorithm converges to the optimal policy at an $O(\frac{1}{t})$ rate. The algorithm is also shown to be extended to an episodic setting. Finally, numerical experiments are conducted to validate the proposed algorithm in the offline setting.

**Strengths:**

- The loss function considered both epistemic and intrinsic uncertainty is novel and the proposed algorithms are shown to achieve convergence to an optimal policy at an $O(\frac{1}{t})$ rate to in the offline setting with infinite data and asymptotic convergence in the episodic setting

**Weaknesses:**

- For the numerical experiments, it would be clearer to show the sub-optimality gap of the optimal policy. While it seams that BR-PG is converging, it's not clear how close it converges to the optimal policy. More details about the implementation of BR-PG should be provided such as choice of step-size (i.e., why it was chosen to be $0.5$) and $r_n$. At the moment, it is difficult to understand what is the differences between the proposed algorithm and it's implementation.

**Questions:**

- Could the definition of the function $L_p$ in Assumption 3.1 be defined?
- In Theorem 4, is it possible to use a decreasing step-size so that the resulting rate is not biased if $r_N$ is finite?
- There are minor formatting error should be fixed. For example, there are missing spaces in Theorem 4:  "Assumption 3.1, 3.2, 3.3,4.1" and Theorem 5: " Assumption 3.1, 3.2, 3.3, 4.1, B.1, B.2 and B.3hold". Moreover, the paper should use the most recent template. The submission has "Under review as a conference paper at ICLR 2024" in the header.
- Minor spelling mistakes:
  - In section 3: "envelop theorem" -> "envelope theorem"
- Clarity
  - In Assumption 3.2, the policy parameter is denoted as $\theta$, but in Eq.3  it is parameterized by $\alpha$.

---

> ### Author Response · Authors · 2024-11-20
>
> **Comment**:
> For the numerical experiments, it would be clearer to show the sub-optimality gap of the optimal policy. While it seams that BR-PG is converging, it's not clear how close it converges to the optimal policy. More details about the implementation of BR-PG should be provided such as choice of step-size (i.e., why it was chosen to be 0.5) and $r_N$. At the moment, it is difficult to understand what is the differences between the proposed algorithm and it's implementation.
>
> **Response**:
> Thank you for your comment and question. For the optimality gap between BR-PG and the optimal policy, please see the table below. The optimal policy is solved by using the value-iteration method. The step-size is chosen based a rough estimation of the Lipschitz constant $L_G$. And $r_n$ is choosen to be $30$. All parameters are now shown. For the last sentence "it is difficult to understand what is the differences between the
> proposed algorithm and it’s implementation.", we would like to seek further clarification from the reviewer. We simply implemented  our proposed algorithm in the numerical experiments, so there is no difference between the implemented version and the proposed algorithm.
>
> | Approach | N=5 |N=50|
> | -------- | ------- |-------- |
> | Approach  | linear loss | linear loss |
> | BR-PG ($\beta = 0$) |33.886 |32.784|
> | BR-PG ($\beta = 0.5$)  | 33.104|32.757|
> |  BR-PG ($\beta = 0.9$)  | 32.854 |32.741|
> |  Optimal Policy   | 32.499 |32.499 |

---

> ### Author Response · Authors · 2024-11-20
>
> **Comment**:
> Could the definition of the function $L_p$ in Assumption 3.1 be defined?
>
> **Response**:
> Thank you for your comment and question. Here $L_p(\Theta,\mu_N)$ is the standard definition of $L_p$ space in real analysis for the set $\Theta$ equipped with probability measure $\mu_N$. Please see  page 65 [1]. Specifically, $L_p(\Theta,\mu_N):= \lbrace f:\int_\Theta |f(\theta)|^p d\mu_N(\theta) <\infty \rbrace $. This assumption requires that  $\int_\Theta |C(\alpha,\theta)|^p d\mu_N(\theta) <\infty$ for all $\alpha$. We will add this explanation for Assumption 3.1.
>
> [1]Walter Rudin. 1987. Real and complex analysis, 3rd ed. McGraw-Hill, Inc., USA.
>
> **Comment**:
>  In Theorem 4, is it possible to use a decreasing step-size so that the resulting rate is not biased if is finite?
>
> **Response**:
> Thank you for your comment and question. We check the proof and we think  it is not possible to use the decreasing step-size. Since $G(\alpha)$ is not  convex due to the parametrization, the proof framework is different from SGD, which uses the decreasing step-size. Using the decreasing step-size $1/\eta_t$ is like replacing $L_G$ with a larger $\eta_t$ in eq.(16) on page 16, which will make the convergence slow.

---

> ### Author Response · Authors · 2024-11-20
>
> **Comment**:
>
> There are minor formatting error should be fixed. For example, there are missing spaces in Theorem 4: "Assumption 3.1, 3.2, 3.3,4.1" and Theorem 5: " Assumption 3.1, 3.2, 3.3, 4.1, B.1, B.2 and B.3hold". Moreover, the paper should use the most recent template. The submission has "Under review as a conference paper at ICLR 2024" in the header.
>
> **Response**:
>
> We thank the reviewer for the careful reading and suggestions. We have fixed these formatting issues in the paper.
>
> **Comment**:
>
> Minor spelling mistakes:
> * In section 3: "envelop theorem" $\rightarrow$ "envelope theorem"
>
> Clarity
> * In Assumption 3.2, the policy parameter   is denoted as $\theta$, but in Eq.3 it is parameterized by $\alpha$.
>
> **Response**:
>
> Thank you for your comment and question. We follow your suggestion and fix these formatting errors. Also, $\theta$ in Assumption 3.2 is the transition kernel parameter, and $\alpha$ in Eq. 3  is the policy parameter.

---

> ### Author Response · Authors · 2024-11-23
>
> We have carefully reviewed your suggestions and uploaded the revised paper, with all changes highlighted in blue for your convenience.
>
> In response to your comments, we have included the numerical results for the optimal policy, added the definition of $L_p$ in Assumption 3, and corrected formatting issues and spelling errors. We hope these revisions address your concerns and enhance the clarity and quality of the paper.

---

> ### Author Response · Authors · 2024-11-25
>
> Dear Reviewer tKe3,
>
> We would be grateful if you could let us know if our responses and revised paper have addressed your questions and concerns. Thank you!

---

> > ### Comment · Reviewer_tKe3 · 2024-11-25
> >
> > Thank you for your response and clarifications. After reading over your thorough responses, I've increased my score to an 6.

---

### Official Review · Reviewer_vKVr · 2024-11-08

**Soundness:** 3
**Presentation:** 3
**Contribution:** 2
**Rating:** 6
**Confidence:** 4

**Summary:**

This paper proposes a novel setting of sequential decision making, where intrinsic uncertainty is handled by considering a generic objective $F(\lambda, P)$ that is convex in the state-action occupancy measure $\lambda$, while epistemic uncertainty is further handled in a Bayesian setting by imposing a risk measure over the posterior distribution of parametrized models $P_{\theta} \sim \mu_N$. The paper then proposes BR-PG, a policy-gradient-based algorithm for the new setting, along with convergence and sample complexity guarantees. The paper also includes numerical simulations that provides a preliminary hint for the applicability of BR-PG in practical settings.

**Strengths:**

1. Most of the proofs in the appendix are checked to be correct. However, some results are cited from other papers that I didn't bother to read or check.
2. The paper proposes a very novel setting that I have never seen before. The setting seems generic to include many existing ones as special cases.
3. The paper is largely well-written. I particularly like the examples (especially the detailed one in Appendix E.1) that provide intuitions about the technical parts.

**Weaknesses:**

1. Although the setting is completely new to me, it remains unclear why we should be interested in this specific setting in the first place. A mixture (a) generic convex objectives, (b) Bayesian setting with risk measures, and (c) both the infinite-horizon and the episodic setting seems too much for an 8-page conference paper, which the authors also (kind of) fail to adequately motivate as a whole. Consequently, the contribution appears to be distracting and a little confusing.
    * Can you provide a detailed real-world setting where the new setting is beneficial? I see "self-driving car" multiple times in the paper, but it would be better to explain why autonomous driving exactly calls for this new modelling, and what's its edge over the classical MDP setting (preferably, via experimental results).
    * What happens if you remove some of the components of the proposed framework? Will the theoretical guarantee be tighter, or even recover some of the known results? How does the performance of the learned policy change (in terms of risk-sensitivity, for example)?
2. The algorithm seem to be not computationally efficient, in the sense that eq. (8) may be hard to solve, and that the posterior distribution $\mu_N$ may be hard to compute or sample from. The authors are urged to discuss how these steps are implemented in practice, and what is the computational complexity of them (even rough arguments of complexity will help here)
    * Since $\mu_N$ can only be approximated, will the performance of the algorithm degrade due to such approximation? Is it possible to reflect such approximation inthe sample complexity bound?
3. The assumptions, though claimed to be "mild/standard" by the authors, are actually a little hard to digest.
    * Assumption 3.2 seems to be restrictive and hard to verify for a generic risk measure. However, I probably shouldn't be too harsh on this point, as two concrete examples are included in the appendix, and more are ready in (Shapiro et. al., 2021).
    * In Theorem 2, it is additionally assumed that $\mu_N$ is a Radon measure, which should have been shown as a lemma if it holds in general (probably under some additional assumptions on $P_{\theta}$ that are *direct* and *mild*). It's weird to assume specific properties of an intermediate variable.
    * Assumption 4.1 seems artificial, or at least I didn't find an intuitive way to understand its physical meaning. Judging from the short proof of Theorem 3, it seems that this assumption only makes technical sense since it's exactly what's needed in the proof.
4. I find it hard to make sense of Theorem 6, since it only guarantees a "constant error bound" that does not converge to 0 as $N$ increases. It appears that having more data doesn't benefit the performance at all, but rather, only adds to the sample complexity of the algorithm. It is also a little surprising to see a sample complexity guarantee with no high-probability statement.
    * Besides, the dependency on $p$, the dimension of the parameter space, is not revealed in the main text. I do see on page 18 that $p$ appears in a factor of $\exp(C/p)$ for some constant $C$, but I'm not sure whether $C$ is positive or negative.
    * As a side note, the notation $L_{i,G}$ vs. $L_{G,i}$ needs to be consistent.
5. In Figure 2, it seems that the losses are roughly the same with $N=5$ and $N=50$ samples. Is this something expected? If so, how should we make sense of it?

**Questions:**

See above.

---

> ### Author Response · Authors · 2024-11-20
>
> **Comment**:
>
> Although the setting is completely new to me, it remains unclear why we should be interested in this specific setting in the first place. A mixture (a) generic convex objectives, (b) Bayesian setting with risk measures, and (c) both the infinite-horizon and the episodic setting seems too much for an 8-page conference paper, which the authors also (kind of) fail to adequately motivate as a whole. Consequently, the contribution appears to be distracting and a little confusing.
>
> *  Can you provide a detailed real-world setting where the new setting is beneficial? I see "self-driving car" multiple times in the paper, but it would be better to explain why autonomous driving exactly calls for this new modelling, and what's its edge over the classical MDP setting (preferably, via experimental results).
>
> **Response**:
>
>
> Thank you for your comment and question. Our paper aims to solve the general objectives and epistemic uncertainty together. For a safe RL problem like autonomous driving, the problem is usually modeled as a constrained MDP. For example, the problem can be modeled as maximizing the total reward about $r$ under the risk-averse safety constraint about $c$:
>   $$\max_\pi \mathbb{E}\left[\sum_{t=0}^{\infty} \gamma^t r\left(s_t, a_t\right) \mid \pi, s_0 \sim \tau\right] $$
> $$\text { s.t. } \rho\left[\sum_{t=0}^{\infty} \gamma^t c\left(s_t, a_t\right) \mid \pi, s_0 \sim \tau\right] \leq D,$$
> where $\rho$ is a risk measure, the total reward can represent the time on the road and the cost can represent the safety risk of different situations. Other risk-averse formulation of autonomous driving is also studied in the literature [1]. It motivates us to study a general objective, which includes, but not limited to, the constrained MDP and risk-averse MDP.
> Another challenge in many problems, such as autonomous driving, is that we may not know the environment completely. The estimation of environment may vary a lot due to the inadequate data.  In this case, classical MDP may work bad due to the large estimation error in a point estimator (such as MLE, maximum likelihood estimator). Instead, our framework uses the Bayesian posterior (a density estimator) to estimate the true unknown parameter. The Bayesian posterior characterizes the likelihood of every point in the parameter space. We further take a risk measure with respect to the posterior, which helps to control the risk due to uncertainty about the true parameter.  Hence, our framework is more robust even in the case of few  data. In numerical results, you can see our method has lower loss than MLE method, which uses the Maximum Likelihood Estimator as a plug-in parameter for the transition kernel and solve this  MDP.
>
> In summary, many problems pose two challenges simultaneously: (1) the objective function is not the classic formulation of a summation of stage-wise costs, and (2) the environment (such as transition probabilities) are not known exactly and have to be estimated from data. We considered both infinite-horizon and episodic setting, because both are commonly considered settings. Moreover, the episodic setting allows a decision maker to incorporate streaming data that come in sequentially in time. We will revise our paper to further clarify the motivation of our problem setting.
>
> [1] Mohammad Naghshvar, Ahmed K. Sadek, Auke J. Wiggers. Risk-averse Behavior Planning for Autonomous Driving under Uncertainty. NeurIPS 2018 Workshop on Machine Learning for Intelligent Transportation Systems.

---

> ### Author Response · Authors · 2024-11-20
>
> **Comment**:
> What happens if you remove some of the components of the proposed framework? Will the theoretical guarantee be tighter, or even recover some of the known results? How does the performance of the learned policy change (in terms of risk-sensitivity, for example)?
>
> **Response**:
>  Thank you for your comment and question. If we remove outer risk measure $\rho$ and assume we know the environment parameter $\theta^c$, then this problem reduces to the convex MDP problem. Our method recovers the result in [1] as we use their variation method as a plug-in method for the inner gradient estimation. It should be noticed that any plug-in method to calculate $\nabla C$ can be used instead of variational method, thus we can recover any convex RL method. Next if we replace the convex general objective $F$ by expectation, then the whole problem reduces to classical MDP. In this case, our method recover the well-known Policy Gradient Theorem in [2], i.e.
> $$\nabla_\alpha G(\alpha)=\sum_{s \in \mathcal{S}} \sum_{a \in \mathcal{A}} Q(s, a)\cdot \nabla_\alpha  \log \pi_\alpha(a \mid s) \lambda(s, a).$$
> For the performance change, take the Conditional Value at Risk (CVaR) with confidence level $\beta$ as an example.  First, for any random variable $X$, Value-at-Risk $VaR_\beta(X)$ is defined as the $\beta$-quantile of $X$, i.e., $VaR_\beta(X):=\inf \{t: \mathbb{P}(X \leq t) \geq \beta\}$, where the confidence level $\beta \in(0,1)$. Then CVaR is defined as $CVaR_\beta(X):=\mathbb{E}\left[X \mid X \geq VaR_\beta(X)\right]$. When $\beta$ is close to $1$,   the policy will be very conservative as $CVaR_\beta(X)$ is similar to the worst-case performance. When $\beta$ is close to $0$, the policy will be robust as $CVaR_\beta(X)$ measures the average performance.
>
> [1] Zhang J, Koppel A, Bedi A S, et al. Variational policy gradient method for reinforcement learning with general utilities[J]. Advances in Neural Information Processing Systems, 2020, 33: 4572-4583.
>
> [2] Sutton R S. Reinforcement learning: An introduction[J]. A Bradford Book, 2018.

---

> ### Author Response · Authors · 2024-11-20
>
> **Comment**:
>
> The algorithm seem to be not computationally efficient, in the sense that eq. (8) may be hard to solve, and that the posterior distribution $\mu_N$ may be hard to compute or sample from. The authors are urged to discuss how these steps are implemented in practice, and what is the computational complexity of them (even rough arguments of complexity will help here).
>
> * Since $\mu_N$ can only be approximated, will the performance of the algorithm degrade due to such approximation? Is it possible to reflect such approximation in the sample complexity bound?
>
> **Reponse**:
>
> Thank you for your comment and question. For Eq.(8),  this min-max problem can be calculated efficiently  by Algorithm 3 in Appendix B.2.1. For some special cases like $F$ is linear or quadratic, we can even analytically compute the optimal solution. For general cases, $O(1/t)$ optimality gap is shown in [1].
>     For computational convenience of the posterior $\mu_N$, we often use conjugate prior in practice, which includes most common cases. Conjugate prior uses a special combination of prior and likelihood function classes such as Gamma-Poisson and Beta-Geometric, and can offer a simple and closed-form expression for posterior parameter update. So it only needs one step to update posterior when using conjugate prior.  In the case there is no conjugate prior available, we have to use some methods such as Bayesian Neural Networks or Markov Chain Monte Carlo.  The performance of the algorithm may be affected as the gradient estimator may be biased. If we can reflect the posterior approximation error in something like the uniform bound of $|\rho_{\theta\sim\mu_N}(C(\alpha,\theta))-\rho_{\theta\sim\tilde{\mu}_N}(C(\alpha,\theta))| $, where $\tilde{\mu}_N$ is an approximation of ${\mu}_N$,  then it will be reflected in sample complexity bound. However, such uniform bound is problem dependent and it is hard to show such bound for general cases.
>
>  [1]Nemirovski A. Prox-method with rate of convergence O (1/t) for variational inequalities with Lipschitz continuous monotone operators and smooth convex-concave saddle point problems[J]. SIAM Journal on Optimization, 2004, 15(1): 229-251.

---

> ### Author Response · Authors · 2024-11-20
>
> **Comment**:
>
> The assumptions, though claimed to be "mild/standard" by the authors, are actually a little hard to digest.
>
> * Assumption 3.2 seems to be restrictive and hard to verify for a generic risk measure. However, I probably shouldn't be too harsh on this point, as two concrete examples are included in the appendix, and more are ready in (Shapiro et. al., 2021).
> * In Theorem 2, it is additionally assumed that $\mu_N$is a Radon measure, which should have been shown as a lemma if it holds in general (probably under some additional assumptions on $P_\theta$ that are direct and mild). It's weird to assume specific properties of an intermediate variable.
> * Assumption 4.1 seems artificial, or at least I didn't find an intuitive way to understand its physical meaning. Judging from the short proof of Theorem 3, it seems that this assumption only makes technical sense since it's exactly what's needed in the proof.
>
> **Response**：
>  Thank you for your comment and question. Assumption 3.2   is  more like a definition since we only focus on this class of coherent risk measures, and most common coherent risk measures we see in other papers satisfy this assumption. We will  change Assumption 3.2 into definition in the revised paper.
>     For Theorem 2 in the  case of continuous parameter space $\Theta$, if the prior is a continuous distribution and the likelihood function is continuous in $\theta$, then the posterior is Radon.  For discrete case, we don't need to care about this assumption. Thus it hold in most cases that we may care about. We will add this explanation about Radon  measure in the footnote.
>     For Assumption 4.1, We will add this justification to the paper "It is hard to show some property of $\xi^*$ in a  general case as the envelop set is given in a general form.  One sufficient condition for Assumption 4.1 to hold is that $\xi^*$ is bounded on  $\Theta$. As  $\Theta$ is a compact and convex set, it is not a strong condition."

---

> ### Author Response · Authors · 2024-11-20
>
> **Comment**:
> I find it hard to make sense of Theorem 6, since it only guarantees a "constant error bound" that does not converge to 0 as $N$ increases. It appears that having more data doesn't benefit the performance at all, but rather, only adds to the sample complexity of the algorithm. It is also a little surprising to see a sample complexity guarantee with no high-probability statement.
> * Besides, the dependency on $p$, the dimension of the parameter space, is not revealed in the main text. I do see on page 18 that $p$ appears in a factor of $\exp(C/p)$ for some constant $C$  , but I'm not sure whether $C$ is positive or negative.
> * As a side note, the notation  $L_{G,i}$ vs $L_{i,G}$ needs to be consistent.
>
> **Response**:
>
> Thank you for your comment and question. In the episodic setting, the total error in each episode is determined by two kinds of error: the estimation error (in the Bayesian posterior due to finite amount of data), and the optimization error (due to finite number of iterations of the gradient optimization algorithm). The  posterior estimation error will converge to $0$ with probability $1$ as the  data size $N$ increases, as shown in Theorem 5. When $N$ increases, we will need less and  less optimization steps to achieve the same bound of the total error , as shown by Theorem 6. So, Theorem 6 shows having more data indeed helps.
> For the reviewer's comment " sample complexity guarantee with no high-probability", we appologize there is a typo here. The constant error bound is for $\mathbb{E}[G_{i}(\alpha_{i,t_{i}})]-G_{i}(\alpha_{i}^*), i=1,\cdots,N$ instead of  $G_{i}(\alpha_{i,t_{i}})-G_{i}(\alpha_{i}^*), i=1,\cdots,N$, which is now consistent with Theorem 4.  For the dependency on  the dimension of transition kernel parameter space $p$, it only appears in the convergence of posteriors in the episodic setting.  In the second formula at page 18, the term $\kappa(\epsilon_2)^{-1/p} e^{-N(\epsilon_1-\epsilon_2)/p}Vol(\Theta)^{1/p}$ shows the effect of $p$, where $ \kappa(\epsilon_2),Vol(\Theta)$ are some volume constants about the parameter space, $N$ is the number of data, and  $\epsilon_1>\epsilon_2$ are two arbitrary small numbers.

---

> ### Author Response · Authors · 2024-11-20
>
> **Comment**:
> In Figure 2, it seems that the losses are roughly the same with $N=5$ and $N=50$ samples. Is this something expected? If so, how should we make sense of it?
>
> **Response**:
> Thank you for your comment and question. There are two differences in the figure of $N=5$ and that of $N=50$. First, the 95\% confidence interval, shown in the shaded band around each curve, is narrower for  $N=50$. As the shaded band for $N=5$ is clear to see, the shaded band for $N=50$ is so narrow that we can nearly only see the curve.  Second, the absolute loss of $N=50$ decreases by about 20\% compared with $N=5$, which is not very clear to see since the large value of MLE enlarges the scale of axis. We will follow
> your suggestion and add a  sentence to state this difference.

---

> ### Author Response · Authors · 2024-11-23
>
> We have carefully considered your suggestions and uploaded the revised paper, with all changes highlighted in blue for your convenience.
>
> In response to your comments, we have provided additional explanations regarding Assumption 3.2, the Radon measure, Assumption 4.1, and Figure 2.  Also, we have fixed the notation errors. We hope these clarifications address your concerns and improve the paper’s overall clarity.

---

> ### Author Response · Authors · 2024-11-25
>
> Dear Reviewer vKVr,
>
> We would be grateful if you could let us know if our responses and revised paper have addressed your questions and concerns. Thank you!

---

> ### Comment · Reviewer_vKVr · 2024-11-25
> **Thanks for the responses.**
>
> I appreciate the authors' positive attitude and dedicated efforts to address my questions and concerns, especially that they have updated the paper to reflect the reviewers' comments. Most of my concerns have been settled to satisfaction, although the assumptions still seem restrictive to some extent.
>
> Considering the overall technical contributions of this paper, I'm raising my rating to 6.

---

### Comment · Area_Chair_QBgM · 2024-12-01
**A question for the authors**

Dear authors,

I have read the paper and your discussion with the reviewers. Congrats to the current good rating.

However, I'm still having concerns about the hidden convexity argument and the corresponding complexity results claimed in this paper. Thanks to the coherent risk structure, one can easily obtain the policy gradient by the min-max variational policy gradient method. However, despite the convexity in $F(\cdot)$, I do not believe the existence of the hidden convexity structure due to one more composition with the risk measure $\rho(\cdot)$.  I guess you might be wrongly using the theory of Zhang et al. (2020).

If I understand correctly, your risk measure is w.r.t. the posterior distribution of the coefficient $\theta$ that characterizes the transition model.  In Zhang et al., the reasoning is the following equivalence and bijection (use naive tabular parameterization):

$$\min_\pi  V(\pi)  \Longleftrightarrow \min_\lambda F(\lambda)  s.t.  A\lambda = b$$

where $A$ and $b$ are constant matrix/vector relying on transition kernel and initial distribution. Then using the bijection between $\pi$ and $\lambda$ one has the 1-1 correspondence between the nonconvex policy optimization and a convex occupancy optimization.

However, for your problem, such bijection does not exist. Consider CVaR (Example 1) with $\Theta = \{\theta_1,\theta_2\}$ and the posterior distribution is uniform, and let $c=0,D=0,l=1$ for simplicity, then the policy optimization problem will be

$$\min_\pi  \min_t t + \frac{1}{1-\alpha}\cdot\frac{\sum_i[d^T\lambda^\pi_{P(\theta_i)}-t]_+}{2} $$

where $\lambda^\pi_{P(\theta_i)}$ is the occupancy measure of policy $\pi$ corresponding to the estimated transition kernel $P(\theta_i)$. Then if one wants to write it as problem of occupancy, then one has to do

$$\min_{\lambda_1,\lambda_2}  \min_t t + \frac{1}{1-\alpha}\cdot\frac{\sum_i[d^T\lambda_i-t]_+}{2}   s.t.  A_1\lambda_1 = b_1, A_2\lambda_2 = b_1$$
Most importantly, to guarantee equivalent to policy optimization problem, they need to ensure $\lambda_1$ and $\lambda_2$ corresponds to the same policy $\pi$, this ends up with an addition constraint that

$$\frac{\lambda_1(s,a)}{\sum_{a'}\lambda_1(s,a')} = \frac{\lambda_2(s,a)}{\sum_{a'}\lambda_2(s,a')} \forall s,a$$

This is not a convex constraint. More importantly, this cannot even be written as a problem of a single $\lambda$ vector in $R^{SA}$. In general, there are two issues.

(1). The bijection may not exist.
(2). The problem is not convex in occupancy measure. This is more fatal, as even if a bijection exists, this still invalidates all the complexity theory of the paper.

Can the authors provide an example and provide the corresponding bijection and the underlying convex problem? (Please use a nontrivial example other than the cases that consider no risk and reduce to standard convex RL in Zhang et al.)

(Otherwise, the authors may only obtain convergence to stationary points, and many sections include motivation and introduction should be rewritten.  see e.g.,
On the convergence and sample efficiency of variance-reduced policy gradient method
J Zhang, C Ni, C Szepesvari, M Wang
Advances in Neural Information Processing Systems, 2021•proceedings.neurips.cc

In this case, it seems no longer important to assume the convexity of F, and the authors may simply consider general F and consider complexity for bounding gradient size)

This will be crucial for us to make the final decision.

Best,

AC

---

> ### Author Response · Authors · 2024-12-03
>
> Thank you for pointing out this issue. The bijection may not be a problem since we only need a local bijecion instead of a global bijection between policy parameter $\alpha$ and occupancy measure. The key problem is that the set of occupancy measure is not convex when we need the occupancy measure under different environmments correspond to the same policy. After carefully checking the proof, there is indeed something wrong with the proof due to the lack of convexity. As you said, we can  only obtain convergence to stationary points. If we want the norm of the gradient for the output policy to  $\le \epsilon$, then we need $\mathcal{O}(\epsilon^{-4})$ sample complexity.  Besides, we only need $F$ to be smooth not convex.
> Thank you for your suggestion, we will improve our paper.

---

### Note · Authors · 2024-12-03

I have read and agree with the venue's withdrawal policy on behalf of myself and my co-authors.